# THE PROBABILISTIC FAULT TOLERANCE OF NEURAL NETWORKS IN THE CONTINUOUS LIMIT

## ABSTRACT

The loss of a few neurons in a brain rarely results in any visible loss of function. However, the insight into what "few" means in this context is unclear. How many random neuron failures will it take to lead to a visible loss of function? In this paper, we address the fundamental question of the impact of the crash of a random subset of neurons on the overall computation of a neural network and the error in the output it produces. We study fault tolerance of neural networks subject to small random neuron/weight crash failures in a probabilistic setting. We give provable guarantees on the robustness of the network to these crashes. Our main contribution is a bound on the error in the output of a network under small random Bernoulli crashes proved by using a Taylor expansion in the continuous limit, where close-by neurons at a layer are similar. The failure mode we adopt in our model is characteristic of neuromorphic hardware, a promising technology to speed up artificial neural networks, as well as of biological networks. We show that our theoretical bounds can be used to compare the fault tolerance of different architectures and to design a regularizer improving the fault tolerance of a given architecture. We design an algorithm achieving fault tolerance using a reasonable number of neurons. In addition to the theoretical proof, we also provide experimental validation of our results and suggest a connection to the generalization capacity problem.

## 1 INTRODUCTION

Understanding the inner working of artificial neural networks (NNs) is currently one of the most pressing questions (20) in learning theory. As of now, neural networks are the backbone of the most successful machine learning solutions (37; 18). They are deployed in safety-critical tasks in which there is little room for mistakes (10; 40). Nevertheless, such issues are regularly reported since attention was brought to the NNs vulnerabilities over the past few years (37; 5; 24; 8).

**Fault tolerance as a part of theoretical NNs research.** Understanding complex systems requires understanding how they can tolerate failures of their components. This has been a particularly fruitful method in systems biology, where the mapping of the full network of metabolite molecules is a computationally quixotic venture. Instead of fully mapping the network, biologists improved their understanding of biological networks by studying the effect of deleting some of their components, one or a few perturbations at a time (7; 12). Biological systems in general are found to be fault tolerant (28), which is thus an important criterion for biological plausibility of mathematical models.

**Neuromorphic hardware (NH).** Current Machine Learning systems are bottlenecked by the underlying computational power (1). One significant improvement over the now prevailing CPU/GPUs is *neuromorphic hardware*. In this paradigm of computation, each neuron is a physical entity (9), and the forward pass is done (theoretically) at the speed of light. However, components of such hardware are small and unreliable, leading to small random perturbations of the weights of the model (41). Thus, robustness to weight faults is an overlooked concrete Artificial Intelligence (AI) safety problem (2). Since we ground the assumptions of our model in the properties of NH and of biological networks, our fundamental theoretical results can be directly applied in these computing paradigms.

**Research on NN fault tolerance.** In the 2000s, the fault tolerance of NNs was a major motivation for studying them (14; 16; 4). In the 1990s, the exploration of microscopic failures was fueled by the hopes of developing neuromorphic hardware (NH) (22; 6; 34). Taylor expansion was one of the tools used for the study of fault tolerance (13; 26). Another line of research proposes sufficient conditions for robustness (33). However, most of these studies are either empirical or are limited to simple architectures (41). In addition, those studies address the worst case (5), which is known to be

more severe than a random perturbation. Recently, fault tolerance was studied experimentally as well. DeepMind proposes to focus on neuron removal (25) to understand NNs. NVIDIA (21) studies error propagation caused by micro-failures in hardware (3). In addition, mathematically similar problems are raised in the study of generalization (29; 30) and robustness (42).

**The quest for guarantees.** Existing NN approaches do not guarantee fault tolerance: they only provide heuristics and evaluate them experimentally. Theoretical papers, in turn, focus on the worst case and not on errors in a probabilistic sense. It is known that there exists a set of small *worst-case* perturbations, *adversarial examples* (5), leading to pessimistic bounds not suitable for the *average case* of random failures, which is the most realistic case for hardware faults. Other branch of theoretical research studies robustness and arrives at error bounds which, unfortunately, scale exponentially with the depth of the network (29). We define the goal of this paper to guarantee that the probability of loss exceeding a threshold is lower than a pre-determined small value. This condition is sensible. For example, self-driving cars are deemed to be safe once their probability of a crash is several orders of magnitude less than of human drivers (40; 15; 36). In addition, current fault tolerant architectures use *mean* as the aggregation of copies of networks to achieve redundancy. This is known to require exponentially more redundancy compared to the *median* approach and, thus, hardware cost. In order to apply this powerful technique and reduce costs, certain conditions need to be satisfied which we will evaluate for neural networks.

**Contributions.** Our main contribution is a theoretical bound on the error in the output of an NN in the case of random neuron crashes obtained in the continuous limit, where close-by neurons compute similar functions. We show that, while the general problem of fault tolerance is NP-hard, realistic assumptions with regard to neuromorphic hardware, and a probabilistic approach to the problem, allow us to apply a Taylor expansion for the vast majority of the cases, as the weight perturbation is small with high probability. In order for the Taylor expansion to work, we assume that a network is smooth enough, introducing the *continuous limit* (39) to prove the properties of NNs: it requires neighboring neurons at each layer to be similar. This makes the moments of the error linear-time computable. To our knowledge, the tightness of the bounds we obtain is a novel result. In turn, the bound allows us to build an algorithm that enhances fault tolerance of neural networks. Our algorithm uses median aggregation which results in only a logarithmic extra cost – a drastic improvement on the initial NP-hardness of the problem. Finally, we show how to apply the bounds to specific architectures and evaluate them experimentally on real-world networks, notably the widely used VGG (38).

**Outline.** In Sections 2-4, we set the formalism, then state our bounds. In Section 5, we present applications of our bounds on characterizing the fault tolerance of different architectures. In Section 6 we present our algorithm for certifying fault tolerance. In Section 7, we present our experimental evaluation. Finally, in Section 8, we discuss the consequences of our findings. Full proofs are available in the supplementary material. Code is provided at the anonymized repo `github.com/iclr-2020-fault-tolerance/code`. We abbreviate Assumption 1 $\to$ A1, Proposition 1 $\to$ P1, Theorem 1 $\to$ T1, Definition 1 $\to$ D1.

## 2 DEFINITIONS OF PROBABILISTIC FAULT TOLERANCE

In this section, we define a fully-connected network and fault tolerance formally.

**Notations.** For any two vectors $x, y \in \mathbb{R}^n$ we use the notation $(x, y) = \sum_{i=1}^{n} x_i y_i$ for the standard scalar product. Matrix $\gamma$-norm for $\gamma = (0, +\infty]$ is defined as $\|A\|_\gamma = \sup_{x \neq 0} \|Ax\|_\gamma / \|x\|_\gamma$. We use the infinity norm $\|x\|_\infty = \max |x_i|$ and the corresponding operator matrix norm. We call a vector $0 \neq x \in \mathbb{R}^n$ $q$-balanced if $\min |x_i| \geq q \max |x_i|$. We denote $[n] = \{1, 2, ..., n\}$. We define the Hessian $H_{ij} = \partial^2 y(x) / \partial x^i \partial x^j$ as a matrix of second derivatives. We write layer indices down and element indices up: $W_l^{ij}$. For the input, we write $x_i \equiv x^i$. If the layer is fixed, we omit its index. We use the element-wise Hadamard product $(x \odot y)_i = x_i y_i$.

**Definition 1.** *(Neural network) A neural network with $L$ layers is a function $y_L \colon \mathbb{R}^{n_0} \to \mathbb{R}^{n_L}$ defined by a tuple $(L, W, B, \varphi)$ with a tuple of weight matrices $W = (W_1, ..., W_L)$ (or their distributions) of size $W_l \colon n_l \times n_{l-1}$, biases $B = (b_1, ..., b_L)$ (or their distributions) of size $b_l \in \mathbb{R}^{n_l}$ by the expression $y_l = \varphi(z_l)$ with pre-activations $z_l = W_l y_{l-1} + b_l$, $l \in [L]$, $y_0 = x$ and $y_L = z_L$. Note that the last layer is linear. We additionally require $\varphi$ to be 1-Lipschitz* [1]. *We assume that the network was trained*

---

[1]1-Lipschitz $\varphi$ s.t. $|\varphi(x) - \varphi(y)| \leqslant |x - y|$. If $\varphi$ is $K$-Lipschitz, we rescale the weights to make $K = 1$: $W_l^{ij} \to W_l^{ij}/K$. This is the general case. Indeed, if we rescale $\varphi(x) \to K\varphi(x)$, then, $y_{l-1} \to Ky'_{l-1}$, and in the sum $z'_l = \sum W^{ij}/K \cdot Ky_{l-1} \equiv z_l$

using input-output pairs $x, y^* \sim X \times Y$ using ERM[2] for a loss $\omega$. Loss layer for input $x$ and the true label $y^*(x)$ is defined as $y_{L+1}(x) = \mathbb{E}_{y^* \sim Y|x} \omega(y_L(x), y^*))$ with $\omega \in [-1, 1]$[3]

**Definition 2.** *(Weight failure) Network* $(L, W, B, \varphi)$ *with weight failures $U$ of distribution $U \sim D|(x, W)$ is the network $(L, W + U, B, \varphi)$ for $U \sim D|(x, W)$. We denote a (random) output of this network as $y^{W+U}(x) = \hat{y}_L(x)$ with activations $\hat{y}_l$ and pre-activations $\hat{z}_l$, as in D1.*

**Definition 3.** *(Bernoulli neuron failures) Bernoulli neuron crash distribution is the distribution with i.i.d. $\xi_l^i \sim \mathrm{Be}(p_l)$, $U_l^{ij} = -\xi_l^i \cdot W_l^{ij}$. For each possible crashing neuron $i$ at layer $l$ we define $U_l^i = \sum_j |U_l^{ij}|$ and $W_l^i = \sum_j |W_l^{ij}|$, the crashed incoming weights and total incoming weights. We note that we see neuron failure as a sub-type of weight failure.*

This definition means that neurons crash independently, and they start to output $0$ when they do. We use this model because it mimics essential properties of NH (41). Components fail relatively independently, as we model faults as random (41). In terms of (41), we consider *stuck-at-0 crashes*, and *passive* fault tolerance in terms of *reliability*.

**Definition 4.** *(Output error for a weight distribution) The error in case of weight failure with distribution $D|(x, W)$ is $\Delta_l(x) = y_l^{W+U}(x) - y_l^W(x)$ for layers $l \in [L+1]$*

We extend the definition of $\varepsilon$-fault tolerance from (23) to the probabilistic case:

**Definition 5.** *(Probabilistic fault tolerance) A network $(L, W, B, \varphi)$ is said to be $(\varepsilon, \delta)$-fault tolerant over an input distribution $(x, y^*) \sim X \times Y$ and a crash distribution $U \sim D|(x, W)$ if $\mathbb{P}_{(x, y^*) \sim X \times Y, U \sim D|(x, W)}\{\Delta_{L+1}(x) \geq \varepsilon\} \leq \delta$. For such network, we write $(W, B) \in \mathrm{FT}(L, \varphi, p, \varepsilon, \delta)$.*

**Interpretation.** To evaluate the fault tolerance of a network, we compute the first moments of $\Delta_{L+1}$. Next, we use tail bounds to guarantee $(\varepsilon, \delta)$-FT. This definition means that with high probability $1 - \delta$ additional loss due to faults does not exceed $\varepsilon$. Expectation over the crashes $U \sim D|x$ can be interpreted in two ways. First, for a large number of neural networks, each having permanent crashes, $\mathbb{E}\Delta$ is the expectation over all instances of a network implemented in the hardware multiple times. For a single network with intermittent crashes, $\mathbb{E}\Delta$ is the output of this one network over repetitions. The recent review study (41) identifies three types of faults: permanent, transient, and intermittent. Our definition 2 thus covers all these cases.

Now that we have a definition of fault tolerance, we show in the next section that the task of certifying or even computing it is hard.

## 3 THE HARDNESS OF FAULT TOLERANCE

In this section, we show why fault tolerance is a hard problem. Not only it is NP-hard in the most general setting but, also, even for small perturbations, the error of the output of can be unacceptable.

### 3.1 NP-HARDNESS

A precise assessment of an NN's fault tolerance should ideally diagnose a network by looking at the outcome of every possible failure, i.e. at the *Forward Propagated Error* (23) resulting from removing every possible subset of neurons. This would lead to an exact assessment, but would be impractical in the face of an exponential explosion of possibilities as by Proposition 1 (proof in the supplementary material).

**Proposition 1.** *The task of evaluating $\mathbb{E}\Delta^k$ for any $k = 1, 2, \ldots$ with constant additive or multiplicative error for a neural network with $\varphi \in C^\infty$, Bernoulli neuron crashes and a constant number of layers is NP-hard.*

We provide a theoretical alternative for the practical case of neuromorphic hardware. We overcome NP-hardness in Section 4 by providing an approximation dependent on the network, and not a constant factor one: for weights $W$ we give $\overline{\Delta}$ and $\underline{\Delta}$ dependent on $W$ such that $\underline{\Delta}(W) \leq \mathbb{E}\Delta \leq \overline{\Delta}(W)$. In addition, we only consider some subclass of all networks.

### 3.2 PESSIMISTIC SPECTRAL BOUNDS

By Definition 4, the fault tolerance assessment requires to consider a weight perturbation $W + U$ given current weights $W$ and the loss change $y_{L+1}(W+U) - y_{L+1}(W)$ caused by it. Mathematically,

---

[2]Empirical Risk Minimization – the standard task $1/k \sum_{k=1}^m \omega(y_L(x_k), y_k^*) \to \min$

[3]The loss is bounded for the proof of Algorithm 1's running time to work

| Quantity | Discrete | | Continuous |
|---|---|---|---|
| Input | $x\colon [n_0] \to \mathbb{R}$ | | $x\colon [0,1] \to \mathbb{R}$ |
| Weights | $W_l\colon [n_l] \times [n_{l-1}] \to \mathbb{R}$ | $\longmapsto$ | $W_l\colon [0,1]^2 \to \mathbb{R}$ |
| Pre-activations | $z_l^i = \sum_j W_l^{ij} y_{l-1}^i + b_l^i$ | | $z_l(t) = \int_0^1 W_l(t,t') y_{l-1}(t') dt' + b_l(t)$ |

Table 1: Correspondence between discrete and continuous quantities. When an regular (discrete) NN is a function mapping vectors to vectors, a continuous NN is an operator mapping functions to functions

this means calculating a local Lipschitz coefficient $K$ (43) connecting $|y_{L+1}(W+U) - y_{L+1}(W)| \leq K|U|$. In the literature, there are known *spectral* bounds on the Lipschitz coefficient for the case of input perturbations. These bounds use the spectral norm of the matrix $\|\cdot\|_2$ and give a global result, valid for any input. This estimate is loose due to its exponential growth in the number of layers, as $\|W\|_2$ is rarely $< 1$. See Proposition 2 for the statement:

**Proposition 2** ($K$ using spectral properties). $\|y_L(x_2) - y_L(x_1)\|_2 \leqslant \|x_2 - x_1\|_2 \cdot \prod_{l=1}^{L} \|W_l\|_2$

The proof can be found in (29) or in the supplementary material. It is also known that high perturbations under small input changes are attainable. Adversarial examples (5) are small changes to the input resulting in a high change in the output. This bound is equal to the one of (23), which is tight in case if the network has the *fewest* neurons. In contrast, in Section 4, we derive our bound in the limit $n \to \infty$.

We have now shown that even evaluating fault tolerance of a given network can be a hard problem. In order to make the analysis practical, we use additional assumptions based on the properties of neuromorphic hardware.

## 4 REALISTIC SIMPLIFYING ASSUMPTIONS FOR NEUROMORPHIC HARDWARE

In this section, we introduce realistic simplifying assumptions grounded in neuromorphic hardware characteristics. We first show that if faults are not too frequent, the weight perturbation would be small. Inspired by this, we then apply a Taylor expansion to the study of the most probable case. [4]

**Assumption 1.** *The probability of failure $p = \max\{p_l | l \in [L]\}$ is small: $p \lesssim 10^{-4}..10^{-3}$*

This assumption is based on the properties of neuromorphic hardware (35). Next, we then use the internal structure of neural networks.

**Assumption 2.** *The number of neurons at each layer $n_l$ is sufficiently big, $n_l \gtrsim 10^2$*

This assumption comes from the properties of state-of-the-art networks (1).

**The best and the worst fault tolerance.** Consider a 1-layer NN with $n = n_0$ and $n_L = n_1 = 1$ at input $x_i = 1$: $y(x) = \sum x_i/n$. We must divide $1/n$ to preserve $y(x)$ as $n$ grows. This is the most robust network, as all neurons are interchangeable. Here $\mathbb{E}\Delta = -p$ and $\mathrm{Var}\,\Delta = p/n$, variance decays with $n$. In contrast, the worst case $y(x) = x_1$ has all but one neuron unused. Therefore $\mathbb{E}\Delta = p$ and $\mathrm{Var}\,\Delta = p$, variance does not decay with $n$.

The next proposition shows that under a mild additional regularity assumption on the network, Assumptions 1 and 2 are sufficient to show that the perturbation of the norm of the weights is small.

**Proposition 3.** *Under A1,2 and if $\{W_l^i\}_{i=1}^{n_l}$ are q-balanced, for $\alpha > p$, the norm of the weight perturbation $U_l^i$ at layer $l$ is probabilistically bounded as: $\delta_0 = \mathbb{P}\{\|U_l^i\|_1 \geq \alpha\|W_l^i\|\} \leq \exp\left(-n_l \cdot q \cdot d_{KL}(\alpha\|p_l)\right)$ with KL-divergence between numbers $a, b \in (0,1)$, $d_{KL}(a,b) = a \log a/b + (1-a) \log (1-a)/(1-b)$ and $W_l^i$ from D3*

Inspired by this result, next, we compute the error $\Delta$ given a small weight perturbation $U$ using a Taylor expansion. [5]

---

[4]The inspiration for splitting the loss calculation into favorable and unfavorable cases comes from (27)

[5]In order to *certify* fault tolerance, we need a precise bounds on the remainder of the Taylor approximation. For example, for ReLU functions, Taylor approximation fails. The supplementary material contains another counter-example to the Taylor expansion of an NN. Instead, we give sufficient conditions for which the Taylor approximation indeed holds.

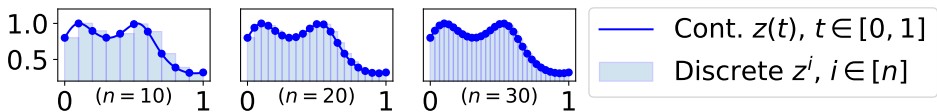

Figure 1: Discrete (standard) neural network approximating a continuous network

**Assumption 3.** *As the width $n$ increases, networks $NN_n$ have a continuous limit (39) $NN_n \to NN_c$, where $NN_c$ is a continuous neural network (19), and $n = \min\{n_l\}$. That network $NN_c$ has globally bounded operator derivatives $D_k$ for orders $k = 1, 2$. We define $D_{12} = \max\{D_1, D_2\}$.[6]*

See Figure 1 for a visualization of A3 and Table 1 for the description of A3. The assumption means that with the increase of $n$, the network uses the same internal structure which just becomes more fine-grained. The continuous limit holds in the case of explicit duplication, convolutional networks and corresponding explicit regularization. The supplementary material contains a more complete explanation.

The derivative bound for order 2 is in contrast to the worse-case spectral bound which would be exponential in depth as in Proposition 2. This is consistent with experimental studies (11) and can be connected to generalization properties via minima sharpness (17).

**Proposition 4.** *Under A3, derivatives are equal the operator derivatives of the continuous limit:*

$$\frac{\partial^k y_L}{\partial y_l^{i^1}...\partial y_l^{i^k}} = \frac{1}{n_l^k}\frac{\delta^k y_L}{\delta y_l(i^1)...\delta y_l(i^k)} + o(1), \ n_l \to \infty$$

For example [7], consider $y(x) = 1/n_1 \sum_{i_1=1}^{n_1} \varphi\left(\sum_{i_0=1}^{n_0} x^{i_0}/n_0\right)$ at $x_i \equiv 1$. Factors $1/n_0$ and $1/n_1$ appear because the network must represent the same $y^*$ as $n_0, n_1 \to \infty$. Then, $\partial y/\partial x_i = \varphi'(1)/n_1$ and $\partial^2 y/\partial x_i \partial x_j = \varphi''(1)/n_1^2$.

**Theorem 1.** *For crashes at layer $l$ and output error $\Delta_L$ at layer $L$ under A1-3 with $q = 1/n_l$ and $r = p + q$, the mean and variance of the error can be approximated as*

$$\mathbb{E}\Delta_L = p_l \sum_{i=1}^{n_l} \left.\frac{\partial y_L}{\partial \xi^i}\right|_{\xi=0} + \Theta_{\pm}(1)D_2 r^2, \ \mathrm{Var}\Delta_L = p_l \sum_{i=1}^{n_l} \left(\left.\frac{\partial y_L}{\partial \xi^i}\right|_{\xi=0}\right)^2 + \Theta_{\pm}(1)D_{12}r^3$$

*By $\Theta_{\pm}(1)$ we denote any function taking values in $[-1, 1]$.[8]*

The full proof of the theorem is in the supplementary material. The remainder terms are small as both $p$ and $q = 1/n_l$ are small quantities under A1-2. In addition, P4 implies $\partial y_L/\partial \xi^i \sim 1/n_l$ and thus, when $n_l \to \infty$, $\mathbb{E}\Delta = \mathcal{O}(1)$ remains constant, and $\mathrm{Var}\Delta_L = \mathcal{O}(1/n_l)$. This is the standard rate in case if we estimate the mean of a random variable by averaging over $n_l$ independent samples, and our previous example in the beginning of the Section shows that it is the best possible rate. Our result shows sufficient conditions under which neural networks allow for such a simplification.[9] In the next sections we use the obtained theoretical evaluation to develop a regularizer increasing fault tolerance, and say which architectures are more fault-tolerant.

## 5 PROBABILISTIC GUARANTEES ON FAULT TOLERANCE USING TAIL BOUNDS

In this section, we apply the results from the previous sections to obtain a probabilistic guarantee on fault tolerance. We identify which kinds of architectures are more fault-tolerant.

---

[6]A necessary condition for $D_k$ to be bounded is to have a reasonable bound on the derivatives of the ground truth function $y^*(x)$. We assume that this function is sufficiently smooth.

[7]The proposition is illustrated in proof-of-concept experiments with explicit regularization in the supplementary material. There are networks for which the conclusion of P4 would not hold, for example, a network with $w^{ij} = 1$. However, such a network does not approximate the same function as $n$ increases since $y(x) \to \infty$, violating A3

[8]The derivative $\partial y_L/\partial \xi^i(\xi) \equiv -\partial y_L(y_l - \xi \odot y_l)/\partial y_l^i \cdot y_l^i$ is interpreted as if $\xi^i$ was a real variable.

[9]However, the dependency $\mathrm{Var}\Delta \sim 1/n_l$ is only valid if $n < p^{-2} \sim 10^8$ to guarantee the first-order term to dominate, $p/n > r^3$. In case if this is not true, we can still render the network more robust by aggregating multiple copies with a mean, instead of adding more neurons. Our current guarantees thus work in case if $p^2 \le n^{-1} \le p$. In the supplementary material, we show that a more tight remainder, depending only on $p/n$, hence decreasing with $n$, is possible. However, it complicates the equation as it requires $D_3$.

Under the assumptions of previous sections, the variance of the error decays as $\mathrm{Var}\Delta \sim \sum C_l p_l/n_l$ as the error superposition is linear (see supplementary material for a proof), with $C_l$ not dependent on $n_l$. Given a fixed budget of neurons, the most fault-tolerant NN has its layers balanced: one layer with too few neurons becomes a single point of failure. Specifically, an optimal architecture with a fixed sum $N = \sum n_l$ has $n_l \sim \sqrt{p_l C_l}$

Given the previous results, certifying $(\varepsilon, \delta)$-fault tolerance is trivial via a Chebyshev tail bound (proof in the supplementary material):

**Proposition 5.** *A neural network under assumptions 1-3 is $(\varepsilon, \delta)$-fault tolerant for $t = \varepsilon - \mathbb{E}\Delta_L > 0$ with $\delta = t^{-2}\mathrm{Var}\Delta_L$ for $\mathbb{E}\Delta$ and $\mathrm{Var}\Delta$ calculated by Theorem 1.*

Evaluation of $\mathbb{E}\Delta$ or $\mathrm{Var}\Delta$ using Theorem 1 would take the same amount of time as one forward pass. However, the exact assessment would need $\mathcal{O}(2^n)$ forward passes by Proposition 1.

In order to make the networks more fault tolerant, we now want to solve the problem of loss minimization under fault tolerance rather than ERM (as previously formulated in (41)): $\inf_{(W,B)\in\mathrm{FT}} \mathcal{L}(w, B)$ where $\mathrm{FT} = \mathrm{FT}(L, \varphi, p, \varepsilon, \delta)$ from Definition 5. Regularizing[10] with Equation 1 can be seen as an approximate solution to the problem above. Indeed, $\mathrm{Var}\Delta \approx p_l \sum_i \left(\frac{\partial L}{\partial y_l^i} \cdot y_l^i\right)^2$ (from T1) is connected to the target probability (P5). Moreover, the network is required to be continuous by A3, which is achieved by making nearby neurons' weights close using a smoothing regularizing function $\mathrm{smooth}(W) \approx \int |W_t'(t, t')| dt dt'$. The $\mu$ term for $q$-balancedness comes from P3 as it is a necessary condition for A3. See the supplementary material for complete details. Here $\hat{\mathcal{L}}$ is the regularized loss, $\mathcal{L}$ the original one, and $\lambda$, $\mu$, $\nu$, $\psi$ are the parameters:

$$\hat{\mathcal{L}}(W) = \mathcal{L}(W) + \lambda \sum_{i=1}^{n_l} \left(\frac{\partial \mathcal{L}}{\partial y^i} \cdot y^i\right)^2 + \mu \left(\frac{\max_i W_l^i}{\min_i W_l^i}\right)^2 + \psi \cdot \mathrm{smooth}(W_l) + \nu \|W\|_\infty \quad (1)$$

We define the terms corresponding to $\lambda, \mu, \psi$ as $R_1 \approx \mathrm{Var}\Delta/p_l$, $R_2 = q^2$, $R_3 = \mathrm{smooth}(W_l)$. If we have achieved $\delta < 1/3$ by P5, we can apply the well-known *median trick* technique (31), drastically increasing fault tolerance. We only use $R$ repetitions of the network with component-wise median aggregation to obtain $(\varepsilon, \delta \cdot \exp(-R))$-fault tolerance guarantee. See supplementary material for the calculations.

In addition, we show that after training, when $\mathbb{E}_x \nabla_W y_{L+1}(x) = 0$, then $\mathbb{E}_x \mathbb{E}_\xi \Delta_{L+1} = 0 + \mathcal{O}(r^2)$ (proof in the supplementary material). This result sheds some light on why neural networks are inherently fault-tolerant in a sense that the mean $\Delta_{L+1}$ is 0. Convolutional networks of architecture `Conv-Activation-Pool` can be seen as a sub-type of fully connected ones, as they just have locally-connected matrices $W_l$, and therefore our techniques still apply. Using large kernel sizes (see supplementary material for discussion), smooth pooling and activations lead to a better approximation.

We developed techniques to assess fault tolerance and to improve it. Now we combine all the results into a single algorithm to certify fault tolerance.

## 6 AN ALGORITHM FOR CERTIFYING FAULT TOLERANCE

We are now in the position to provide an algorithm (Algorithm 1) allowing to reach the desired $(\varepsilon, \delta)$-fault tolerance via training with our regularizer and then physically duplicating the network a logarithmic amount of times in hardware, assuming independent faults. We note that our algorithm works for a single input $x$ but is easily extensible if the expressions in Propositions are replaced with expectations over inputs (see supplementary material).

In order to estimate the required number of neurons, we use bounds from T1 and P5 which require $n \sim p/\varepsilon^2$. However, using the median approach allows for a fast exponential decrease in failure probability. Once the threshold of failing with probability $1/3$ is reached by P5, it becomes easy to reach *any* required guarantee. The time complexity (compared to the one of training) of the algorithm is $\mathcal{O}(D_{12} + C_l p_l/\varepsilon^2)$ and space complexity is equal to that of one training call. See supplementary material for the proofs of resource requirements and correctness.

---

[10]We note that the gradient of $\mathrm{Var}\Delta$ is linear time-computable since it is a Hessian-vector product.

[11]More neurons do not solve the problem, as $\mathbb{E}\Delta$ stays constant with the growth of $n$ by Theorem 1. Intuitively, this is due to the fact that if a mean of a random variable is too high, more repetitions do not make the estimate lower.

**Data:** Dataset $D$, input-output point $(x, y^*)$, failure probabilities $p_l$, depth $L$, activation function $\varphi \in C^\infty$, target $\varepsilon$ and $\delta'$, the error tolerance parameters from the Definition 5, maximal complexity guess $C \approx \int |y'_l(t)|dt \approx R_3^{guess}$

**Result:** An architecture with $(\varepsilon, \delta')$-fault-tolerance on $x$

1   Select initial width $N = (n_1, ..., n_{L-1})$;

2   **while** *true* **do**

3      Train a network to obtain $W, B$;

4      Compute $q$ from Proposition 3;

5      **If** $q < 10^{-2}$, increase regularization parameter $\mu$ from Eq. 1, **continue**; *// go to line 3*;

6      Compute $\delta_0$ from Proposition 3 using $q$;

7      **If** $\delta_0 > 1/3$, increase $n$ by a constant amount, **continue**;

8      Compute $R_3$ from Eq. 1;

9      **If** $R_3 > C$, increase regularization parameter $\psi$ from Eq. 1, **continue**;

10      Compute $\mathbb{E}\Delta$ and $\text{Var}\Delta$ from Theorem 1;

11      **If** $\mathbb{E}\Delta > \varepsilon$, **output** infeasible; *// cannot do better than the mean[11]*;

12      Compute $\delta$ from Proposition 5;

13      **If** $\delta > 1/3$, increase $n$ by a constant amount and increase $\lambda$ in Eq. 1, **continue**;

14      Compute $R = \mathcal{O}(\log \frac{1}{\delta'})$;

15      **Output** number of repetitions $R$, layer widths $N$, parameters $W, B$;

16   **end**

**Algorithm 1:** Achieving fault tolerance after training. The numbers $q_{\max} = 10^{-2}$ and $\delta_{\max} = 1/3$ are chosen for simplicity of proofs. The asymptotic behavior does not change with different numbers, as long as $\delta_{\max} < 1/2$ and the constraints on $q$ mentioned in the supplementary material are met

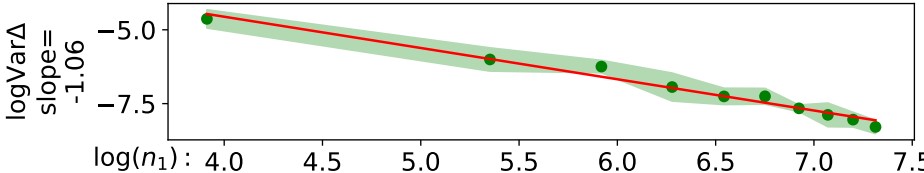

Figure 2: The effect of the layer width $n_l$, $l = 1$, horizontal axis on the variance of the fault tolerance error $\text{Var}\Delta$, vertical axis

## 7   EXPERIMENTAL EVALUATION

In this section, we test the theory developed in previous sections in proof-of-concept experiments. We first show that we can correctly estimate the first moments of the fault tolerance using T1 for small (10-50 neurons) and larger networks (VGG). We test the predictions of our theory such as decay of $\text{Var}\Delta$, the effect of our regularizer and the guarantee from Algorithm 1. See the supplementary material for the technical details where we validate the assumption of derivative decay (A3) explicitly. Our code is provided at the anonymized repository `github.com/iclr-2020-fault-tolerance/code`.

**Increasing training dropout.** We train sigmoid networks with $N \sim 100$ on MNIST (see `ComparisonIncreasingDropoutMNIST.ipynb`). We use probabilities of failure at inference and training stages $p_i = 0.05$ at the first layer and 10 values of $p_t \in [0, 1.2p_i]$. The experiment is repeated 10 times. When estimating the error experimentally, we choose 6 repetitions of the training dataset to ensure that the variance of the estimate is low. The results are in the Table 2. The experiments show that crashing MAE (Mean Absolute Error for the network with crashes at inference) is most dramatically affected by dropout. Specifically, training with $p_t \sim p_i$ makes network more robust at inference, which was well-established before. Moreover, the bound from T1 can correctly order which network is trained with bigger dropout parameter with only $4\%$ rank loss, which is the fraction of incorrectly ordered pairs. All other quantities, including norms of the weights, are not able to order networks correctly. See supplementary material for a complete list of metrics in the experiment.

| Quantity | Train rank loss | Test rank loss |
|---|---|---|
| Crashing, MAE | **5.6%** | **5.6%** |
| Crashing, Accuracy | 19.8% | 17.7% |
| Correct, MAE | 23.3% | 22.0% |
| Correct, Accuracy | 31.7% | 39.9% |

(a) Experimental metrics

| Quantity | Rank Loss |
|---|---|
| T1 Var$\Delta$ | **3.6%** |
| P2 $\mathbb{E}\Delta$ | 24.8% |
| P2 Var$\Delta$ | 31.7% |
| T1 $\mathbb{E}\Delta$ | 40.8% |

(b) Theoretical bounds

Table 2: Comparison of networks trained with increased dropout. Rank loss between $p_{train}$ and various metrics. Experiment shows crashing and correct networks, MAE or accuracy and test/train datasets. Theoretical bounds include P2[12] and T1.

**Regularization for fault tolerance**. Previously, our bound is demonstrated to be able to correctly predict which network is more resilient. We therefore use it as a regularization technique suggested by Eq. 1, see `Regularization.ipynb`. We establish that the resilience of the network regularized with Dropout is similar to that of a network regularized with the bound

**Testing the bound on larger networks.** We test the bound on VGG16 and on a smaller convnet, see `ConvNetTest-MNIST.ipynb` and `ConvNetTest-VGG16.ipynb` and verify that they correctly predict the magnitude of the error

**Architecture and fault tolerance.** Comparing different architectures on a single image with $p = 0.01$ (VGG16, VGG19, MobileNet) shows (and `ConvNetTest-ft.ipynb`) that the bigger the mean width of the layer (approximated by the number of parameters), the better is the fault tolerance, as predicted in Section 5. In addition, training networks on the MNIST dataset (see `FaultTolerance-Continuity-FC-MNIST.ipynb`) shows a decrease in variance with $n_l$ as predicted by Theorem 1, see Figure 2: the variance decays as $1/n_l$. We regularize with $\psi = (10^{-4}, 10^{-2})$ for derivatives and smoothing respectively (see supplementary material for explanation of coefficients) and $\lambda = 0.001$.

**Testing the algorithm.** We test the Algorithm 1 on the MNIST dataset for $\varepsilon = 9 \cdot 10^{-3}$, $\delta = 10^{-5}$ and obtain $R = 20$, $n_1 = 500$, $\lambda = 10^{-6}$, $\mu = 10^{-10}$, $\psi = (10^{-4}, 10^{-2})$. We evaluate the tail bound experimentally. Our experiment demonstrates the guarantee given by Proposition 5 and can be seen as an experimental confirmation of the algorithm's correctness. See `TheAlgorithm.ipynb`.

We hence conclude that our proof-of-concept experiments show an overall validity of our assumptions and of our approach.

## 8 CONCLUSION

Fault tolerance is an important overlooked concrete AI safety issue (2). This paper describes a probabilistic fault tolerance framework for NNs that allows to get around the NP-hardness of the problem. Since the crash probability in neuromorphic hardware is low, we can simplify the problem to allow for a polynomial computation time. We use the tail bounds to motivate the assumption that the weight perturbation is small. This allows us to use a Taylor expansion to compute the error. To bound the remainder, we require sufficient smoothness of the network, for which we use the continuous limit: nearby neurons compute similar things. After we transform the expansion into a tail bound to give a bound on the loss of the network. This gives a probabilistic guarantee of fault tolerance. Using the framework, we are able to guarantee sufficient fault tolerance of a neural network given parameters of the crash distribution. We then analyze the obtained expressions to compare fault tolerance between architectures and optimize for fault tolerance of one architecture. We test our findings experimentally on small networks (MNIST) as well as on larger ones (VGG-16, MobileNet). Using our framework, one is able to deploy safer networks into neuromorphic hardware.

Mathematically, the problem that we consider is connected to the problem of generalization (29; 27) since the latter also considers the expected loss change under a small random perturbation $\mathbb{E}_{W+U}\mathcal{L}(W + U) - \mathcal{L}(W)$, except that these papers consider Gaussian noise and we consider Bernoulli noise. Evidence (32), however, shows that sometimes networks that generalize well are not necessarily fault-tolerant. Since the tools we develop for the study of fault tolerance could as well be applied in the context of generalization, they could be used to clarify this matter.

---

[12]Variance for P2 is derived in the supplementary material

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

# Probabilistic Fault Tolerance
# of Neural Networks
# in the Continuous Limit.
# Supplementary material

Anonymous authors

September 2019



## 1 Introduction

First we prove all the propositions (labeled Proposition 1, 2, ...) from the main paper. Their names and sections match the main paper. We also give additional results, they are labelled Additional Proposition 1, 2, etc. If they are formal statements of the results referred to in the main paper, they are in the same section as the reference. We abbreviate Assumption 1 $\to$ A1, Proposition 1 $\to$ P1, Theorem 1 $\to$ T1, Definition 1 $\to$ D1.

  Less precise statements with possible future research directions on fundamental questions required to make the guarantee even more strong are flushed right.

## 2 Definition of Probabilistic Fault Tolerance

**Notations.** For any two vectors $x, y \in \mathbb{R}^n$ we use the notation $(x, y) = \sum_{i=1}^n x_i y_i$ for the standard scalar product. Matrix $\gamma$-norm for $\gamma = (0, +\infty]$ is defined as $\|A\|_\gamma = \sup_{x \neq 0} \|Ax\|_\gamma / \|x\|_\gamma$. We use the infinity norm $\|x\|_\infty = \max |x_i|$ and the corresponding operator matrix norm. We call a vector $0 \neq x \in \mathbb{R}^n$ $q$-balanced if $\min |x_i| \geq q \max |x_i|$. We denote $[n] = \{1, 2, ..., n\}$. We define the Hessian $H_{ij} = \partial^2 y(x) / \partial x^i \partial x^j$ as a matrix of second derivatives. We write layer indices down and element indices up: $W_l^{ij}$. For the input, we write $x_i \equiv x^i$. If the layer is fixed, we omit its index. We use the element-wise Hadamard product $(x \odot y)_i = x_i y_i$.

**Definition 1.** *(Neural network) A neural network with $L$ layers is a function $y_L \colon \mathbb{R}^{n_0} \to \mathbb{R}^{n_L}$ defined by a tuple $(L, W, B, \varphi)$ with a tuple of weight matrices $W = (W_1, ..., W_L)$ (or their distributions) of size $W_l \colon n_l \times n_{l-1}$, biases $B = (b_1, ..., b_L)$ (or their distributions) of size $b_l \in \mathbb{R}^{n_l}$ by the expression $y_l = \varphi(z_l)$ with pre-activations $z_l = W_l y_{l-1} + b_l$, $l \in [L]$, $y_0 = x$ and $y_L = z_L$. Note that the last layer is linear. We additionally require $\varphi$ to be 1-Lipschitz [1]. We assume that the network was trained using input-output pairs $x, y^* \sim X \times Y$ using ERM[2] for a loss $\omega$. Loss layer for input $x$ and the true label $y^*(x)$ is defined as $y_{L+1}(x) = \mathbb{E}_{y^* \sim Y|x} \omega(y_L(x), y^*))$ with $\omega \in [-1, 1]$[3]*

**Definition 2.** *(Weight failure) Network $(L, W, B, \varphi)$ with weight failures $U$ of distribution $U \sim D|(x, W)$ is the network $(L, W + U, B, \varphi)$ for $U \sim D|(x, W)$. We denote a (random) output of this network as $y^{W+U}(x) = \hat{y}_L(x)$ with activations $\hat{y}_l$ and pre-activations $\hat{z}_l$, as in D1.*

---

[1] 1-Lipschitz $\varphi$ s.t. $|\varphi(x) - \varphi(y)| \leqslant |x - y|$. If $\varphi$ is $K$-Lipschitz, we rescale the weights to make $K = 1$: $W_l^{ij} \to W_l^{ij}/K$. This is the general case. Indeed, if we rescale $\varphi(x) \to K\varphi(x)$, then, $y_{l-1} \to Ky'_{l-1}$, and in the sum $z'_l = \sum W^{ij}/K \cdot Ky_{l-1} \equiv z_l$

[2] Empirical Risk Minimization – the standard task $1/k \sum_{k=1}^m \omega(y_L(x_k), y_k^*) \to \min$

[3] The loss is bounded for the proof of Algorithm 1's running time to work

**Definition 3.** *(Bernoulli neuron failures) Bernoulli neuron crash distribution is the distribution with i.i.d. $\xi_l^i \sim$ Be($p_l$), $U_l^{ij} = -\xi_l^i \cdot W_l^{ij}$. For each possible crashing neuron $i$ at layer $l$ we define $U_l^i = \sum_j |U_l^{ij}|$ and $W_l^i = \sum_j |W_l^{ij}|$, the crashed incoming weights and total incoming weights. We note that we see neuron failure as a sub-type of weight failure.*

**Definition 4.** *(Output error for a weight distribution) The error in case of weight failure with distribution $D|(x, W)$ is $\Delta_l(x) = y_l^{W+U}(x) - y_l^{W}(x)$ for layers $l \in [L+1]$*

We extend the definition of $\varepsilon$-fault tolerance from [18] to the probabilistic case:

**Definition 5.** *(Probabilistic fault tolerance) A network $(L, W, B, \varphi)$ is said to be $(\varepsilon, \delta)$-fault tolerant over an input distribution $(x, y^*) \sim X \times Y$ and a crash distribution $U \sim D|(x, W)$ if $\mathbb{P}_{(x,y^*)\sim X \times Y, U \sim D|(x,W)}\{\Delta_{L+1}(x) \geq \varepsilon\} \leq \delta$. For such network, we write $(W, B) \in \mathrm{FT}(L, \varphi, p, \varepsilon, \delta)$.*

# 3 The Hardness of The Fault Tolerance

## 3.1 NP-Hardness

**Proposition 1.** *The task of evaluating $\mathbb{E}\Delta^k$ for any $k = 1, 2, ...$ with constant additive or multiplicative error for a neural network with $\varphi \in C^\infty$, Bernoulli neuron crashes and a constant number of layers is NP-hard.*

*Proof.* To prove that a problem is NP-hard, it suffices to take another NP-hard problem, and reduce any instance of that problem to our problem, meaning that solving our problem would solve the original one.

We take the NP-hard Subset Sum problem. It states: given a finite set of integers $x_i \in \mathbb{Z}$, $i \in [M]$, determine if there exist a non-empty subset $S \subseteq [M]$ such that $\sum_{x \in S} x_i = 0$.

We take a subset sum instance $x_i \in \mathbb{Z}$ and feed it as an input to a neural network with two first layer neurons with $\varphi$ being a piecewise-linear function from 0 to 1 at points $-\varepsilon$ and $\varepsilon$ for some fixed $\varepsilon \in (0, 1)$. Note that in this proof we only compute $\varphi$ at integer points, and therefore it is possible re-define $\varphi \in C^\infty$ such that it has same values in natural points.

Note that inputs to it are integers, therefore the outputs are 1 if and only if (iff) the sum is greater than zero. First neuron has coefficients all 1s and second all $-1$s with no bias. The next neuron has coefficients 1 and 1 for both inputs, a bias $-1.5$ and threshold activation function, outputting 1 only if both inputs are 1. Again, since inputs to this neuron are integers, we can re-define $\varphi$ to be $C^\infty$. The final neuron is a (linear) identity function to satisfy Definition 1. It takes the previous neuron as its only input. Now, we see that $y(x) = 1$ if and only if the sum of inputs to a network is 0.

We now feed the entire set $S$ to the network as its input $x$. In case if $y(x) = 1$ (which is easy to check), we have arrived at a solution, as the whole set has zero sum. In the following we consider the case when $y(x) = 0$.

Suppose that there exist an algorithm calculating answer to the question $\mathbb{E}\Delta^k > z$ for any finite-precision $z$. Now, the expectation has terms inside $\mathbb{E}\Delta^k = \sum_{s \in S} p^{|s|}(1-p)^{1-|s|}y^k(x \odot s)$. Suppose that one of the terms is non-zero. Then $y(x \odot s) \neq 0$, which means that the threshold neuron outputs a value $> 0$ which means that sum of its inputs is greater than 1.5. This can only happen if they are both 1, which means that sum for a particular subset is both $\geq 0$ and $\leq 0$. By setting $z = 0$ we solve the subset sum problem using one call to an algorithm determining if $\mathbb{E}\Delta^k > 0$. Indeed, in case if the original problem has no solution, the algorithm will output $\mathbb{E}\Delta^k \leq 0$ as there are no non-zero terms inside the sum. In case if there is a solution, there will be a non-zero term in the sum and thus $\mathbb{E}\Delta^k > 0$ (all terms are non-negative).

Now, suppose there exist an algorithm which always outputs an additive approximation to $\mathbb{E}\Delta^k$ giving numbers $\mu$ and $\varepsilon$ such that $\mathbb{E}\Delta^k \in (\mu - \varepsilon, \mu + \varepsilon)$ for some constant $\varepsilon$. Take a network from previous example with additional layer scaling output by $C = \frac{2\varepsilon}{p^M}$. Take a subset sum instance $x_i$. Output YES iff $0 \notin (\mu - \varepsilon, \mu + \varepsilon)$. This is correct because if $0 \notin (\mu - \varepsilon, \mu + \varepsilon)$, then it must be that $\mathbb{E}\Delta^k > 0$ and vice versa. Multiplicative approximation has an analogous proof where we scale the outputs even more. $\square$

We note that computing the distribution of $\Delta$ for one neuron, binary input and weights and a threshold activation function is known as *noise sensitivity* in theoretical Computer Science. There exists an exact assessment for $W_1^{1i} \equiv w^i = 1$[4], however for the case $w^i \neq 1$ the exact distribution is unknown [6].

## 3.2 Pessimistic Spectral Bounds

**Additional Proposition 1** (Norm bound or bound b1)**.** *For any norm $\|\cdot\|$ the error $\Delta_L$ at the last layer on input $x$ and failures in input can be upper-bounded as (for $\xi \odot x$ being the crashed input)*

$$\|\Delta_L\| \leqslant \|W_L\| \cdot \ldots \cdot \|W_1\| \|\xi \odot x - x\|$$

*Proof.* We assume a faulty input (crashes at layer 0). By Definition 4 the error at the last layer is $\Delta_L = \hat{y}_L - y_L$. By definition 1, $y_L = W_L y_{L-1} + b_L$ and by Definition 3, $\hat{y}_L = W_L \hat{y}_{L-1} + b_L$. Thus, $\|\Delta_L\| = \|W_L \hat{y}_{L-1} + b_L - W_L y_{L-1} - b_L\| = \|W_L(\hat{y}_{L-1} - y_{L-1})\| \leq \|W_L\| \|\hat{y}_{L-1} - y_{L-1}\|$.

Next, since $\hat{y}_{L-1} = \varphi(W_{L-1}\hat{y}_{L-2} + b_{L-1})$ and $y_{L-1} = \varphi(W_{L-1}y_{L-2} + b_{L-1})$, we use the 1-Lipschitzness of $\varphi$. In particular, we use that for two vectors $x$ and $y$ and $\varphi(x), \varphi(y)$ applied element-wise, we have $\|\varphi(x) - \varphi(y)\| \leq \|x - y\|$ because the absolute difference in each component is less on the left-hand side. We thus get $\|\hat{y}_{L-1} - y_{L-1}\| \leq \|W_{L-1}\hat{y}_{L-2} + b_{L-1} - W_{L-1}\hat{y}_{L-2} - b_{L-1}\| = \|W_{L-1}(\hat{y}_{L-2} - y_{L-2})\| \leq \|W_{L-1}\| \|\hat{y}_{L-2} - y_{L-2}\|$. Plugging it in to the previous equation, we have $\|\Delta_L\| \leq \|W_L\| \cdot \|W_{L-1}\| \|\hat{y}_{L-2} - y_{L-2}\|$

Now, since the inner layer act in a similar manner, we inductively apply the same argument for underlying layers and get

$$\|\Delta_L\| \leq \|W_L\| \cdot \ldots \cdot \|W_1\| \|\hat{x} - x\|$$

Moreover we have failing input, thus $\hat{x} = \xi \odot x$ which completes the proof. $\qquad\square$

**Proposition 2** (*K* using spectral properties [21])**.** $\|y_L(x_2) - y_L(x_1)\|_2 \leqslant \|x_2 - x_1\| \cdot \prod_{l=1}^{L} \|W_l\|_2$

*Proof.* Application of AP1 for $\|\cdot\| = \|\cdot\|_2$ and $x_2 = \xi \odot x$ $\qquad\square$

We see, there are bounds considering different norms of matrices $\|A\|_\gamma$ (AP1) or using the triangle inequality and absolute values (AP2). They still lead to pessimistic estimates. All these bounds are known to be loose [19] due to $\|W\| = \sup_{x \neq 0} \|Wx\|/\|x\|$ being larger than the input-specific $\|Wx\|/\|x\|$. We will circumvent this issue by considering the *average* case instead of the worst case.

**Additional Corollary 1** (Infinity norm, connecting [9] to norm bounds)**.** *For an input $x$ with $C = \|x\|_\infty$ for failures at the input with $\mathcal{O}(1)$ dependent only on layer size, but not on the weights,*

$$\|\Delta_L\|_\infty \leqslant pC\|W_L\|_\infty \cdot \ldots \cdot \|W_1\|_\infty \cdot \mathcal{O}(1) + \mathcal{O}(p^2)$$

*Here $\|x\|_\infty = \max |x_i|$*

*Proof.* First we examine the expression (4) from the other paper [9] and show that it is equivalent to the result we are proving now:

$$Erf = \mathbb{E}\|\Delta\|_\infty \leq \sum_{l=1}^{L} C_l f_l K^{L-l} w_m^{(L)} \prod_{l'=l+1}^{L} (N_{l'} - f_{l'}) w_m^{(l')}$$

here we have $C_l$ the maximal value at layer $l$, $K$ the Lipschitz constant, $w_m$ is the maximal over output neurons (rows of $W$) and mean absolute value over input neurons (columns of $W$) weight, $f_l$ is the number of crashed neurons.

Now we set $f_1 = pN_1$ and $f_i = 0$ for $i > 1$ and moreover we assume $K = 1$ as in the main paper.

---

[4]As we only have one neuron, the index is one-dimensional

Therefore the bound is rewritten as:

$$\mathbb{E}\|\Delta\|_\infty \leq C_1 N_1 p w_m^{(L)} \prod_{l'=2}^{L} N_{l'} w_m^{(l')}$$

Now we notice that the quantity $N_l w_m^l = \|W^l\|_\infty$ and therefore

$$\mathbb{E}\|\Delta\|_\infty \leq p C_1 \frac{N_1}{N_L} \|W_2\|_\infty \cdot ... \|W_L\|_\infty$$

Now we assume that the network has one more layer so that the bound from [9] works for a faulty input in the original network:

$$\mathbb{E}\|\Delta\|_\infty \leq p C \frac{N_0}{N_L} \|W_1\|_\infty \cdot ... \cdot \|W_L\|_\infty$$

Here $C = \max\{|x_i|\} = \|x\|_\infty$. Next we prove that result independently using Additional Proposition 1 for $\|\cdot\|_\infty$

$$\|\Delta_L\|_\infty \leq \|W_L\|_\infty \cdot ... \cdot \|W_1\|_\infty \|\xi \odot x - x\|_\infty$$

Now we calculate $\mathbb{E}\|\xi \odot x - x\|_\infty$. We write the definition of the expectation with $f(p, n, k) = p^k (1-p)^{n-k} = p^k + \mathcal{O}(p^{k+1})$ being the probability that a binary string of length $n$ has a particular configuration with $k$ ones, if its entries are i.i.d. Bernoulli $Be(p)$. Here $S_l$ is the set of all possible network crash configurations at layer $l$. Each configuration $s_l \in S_l$ describes which neurons are crashed and which are working. We have $|S_l| = 2^{n_l}$.

$$\mathbb{E}\|\xi \odot x - x\|_\infty = \sum_{s \in S} f(p, n, |s|) \max\{s \odot x - x\}$$

Now since for $|s| = 0$ the $\max\{s \odot x - x\} = \max\{x - x\} = 0$, we consider cases $|s| = 1$ and $|s| > 1$. For $|s| > 1$ the quantity $f(p, n, |s|) = \mathcal{O}(p^2)$ and therefore, in the first order:

$$\mathbb{E}\|\xi \odot x - x\|_\infty = p \sum_i |x_i| + \mathcal{O}(p^2) \leq p N_0 \|x\|_\infty + \mathcal{O}(p^2)$$

Next we plug that back into the expression for $\mathbb{E}\|\Delta\|_\infty$:

$$\mathbb{E}\|\Delta\|_\infty \leq \|W_L\|_\infty \cdot ... \cdot \|W_1\|_\infty p N_0 \|x\|_\infty + \mathcal{O}(p^2)$$

Now we note that this expression and the expression from [9] differ only in a numerical constant in front of the bound: $\frac{N_0}{N_L}$ instead of $N_0$, but the bounds behave in the same way with respect to the weights.

$\square$

**Additional Proposition 2** (Absolute value bound or bound b2). *The error on input $x$ can be upper-bounded as:*

$$\mathbb{E}|\Delta_L| \leqslant p |W_L| \cdot ... \cdot |W_1| |x|$$

*For $|W|$ being the matrix of absolute values $(|W|)_{ij} = |W_{ij}|$. $|x|$ means component-wise absolute values of the vector.*

*Proof.* This expression involves absolute value of the matrices multiplied together as matrices and then multiplied by an absolute value of the column vector. The absolute value of a column vector is a vector of element-wise absolute values.

We assume a faulty input. By Definition 4, the error at the last layer is $\Delta_L = \hat{y}_L - y_L$. By definition 1, $y_L = W_L y_{L-1} + b_L$ and by Definition 3, $\hat{y}_L = W_L \hat{y}_{L-1} + b_L$. Thus for the $i$'th component of the error,

$$|\Delta_L^i| = |W_L^i \hat{y}_{L-1} + b_L^i - W_L^i y_{L-1} - b_L^i| = |W_L^i(\hat{y}_{L-1} - y_{L-1})| = |\sum_j W^{ij}(\hat{y}_{L-1}^j - y_{L-1}^j)|$$

By the triangle inequality,

$$|\Delta_L^i| \le \sum_j |W_L^{ij}| \cdot |\hat{y}_{L-1}^j - y_{L-1}^j| = (|W_L||\hat{y}_{L-1} - y_{L-1}|)^i$$

Next we go one level deeper according to Definition 1:

$$|\hat{y}_{L-1}^j - y_{L-1}^j| = |\varphi(W_{L-1}^j \hat{y}_{L-2} + b_{L-1}^j) - \varphi(W_{L-1}^j y_{L-2} + b_{L-1}^j)|$$

And then apply the 1-Lipschitzness property of $\varphi$:

$$|\hat{y}_{L-1}^j - y_{L-1}^j| \le |W_{L-1}^j \hat{y}_{L-2} + b_{L-1}^j - W_{L-1}^j y_{L-2} - b_{L-1}^j| = |W_{L-1}^j (\hat{y}_{L-2} - y_{L-2})|$$

This brings us to the previous case and thus we analogously have

$$|\Delta_L| \le |W_L| \cdot |W_{L-1}| \cdot |\hat{y}_{L-2} - y_{L-2}|$$

Inductively repeating these steps, we obtain:

$$|\Delta_L| \le |W_L| \cdot \ldots \cdot |W_1||\hat{x} - x|$$

Now we take the expectation and move it inside the matrix product by linearity of expectation:

$$\mathbb{E}|\Delta_L| \le \mathbb{E}|W_L| \cdot \ldots \cdot |W_1||\hat{x} - x| = |W_L| \cdot \ldots \cdot |W_1|\mathbb{E}|\hat{x} - x|$$

The last expression involves $\mathbb{E}|\hat{x} - x|$. This is component-wise expectation of a vector and we examine a single component. Since $\xi^i$ is a Bernoulli random variable,

$$\mathbb{E}|x_i \xi^i - x_i| = p \cdot |x_i| + (1 - p) \cdot 0$$

Plugging it into the last expression for $\mathbb{E}|\Delta_L|$ proves the proposition. $\qquad\square$

# 4 Realistic Simplifying Assumptions for Neuromorphic Hardware

## 4.1 When is The Weight Perturbation Small?

In this section we give sufficient conditions for which case the probability of large weight perturbation under a crash distribution is small. First, we define the properties of neuromorphic hardware.

**Assumption 1.** *The probability of failure $p = \max\{p_l | l \in [L]\}$ is small: $p \lesssim 10^{-4}..10^{-3}$*

**Assumption 2.** *The number of neurons at each layer $n_l$ is sufficiently big, $n_l \gtrsim 10^2$*

**Additional Assumption 1.** *Vectors $W_l^i = \sum_j |W_l^{ij}|$ are q-balanced for $q > 0$ for each layer $l$.[5]*

---

[5] A similar assumption on an "even distribution" of weights was made in [20].

129  **Toy examples.**  Naturally, we expect the error to decay with an increase in number of neurons $n$, because of
130  redundancy. We show that this might not be always the case. First, consider a 1-layer NN with $n = n_0$ and
131  $n_L = n_1 = 1$ at input $x_i = 1$: $y(x) = \sum x_i/n$. This is most robust network, as all neurons are interchangeable as
132  they are computing the same function each. Essentially, this is an estimate of the mean of $\xi_i$ given $n$ samples. Here
133  $\mathbb{E}\Delta = -\sum \mathbb{E}\xi_i x_i = -p$ and $\text{Var}\Delta = 1/n \sum \text{Var}\xi_i = p(1-p)/n \sim p/n$, variance decays with $n$. In contrast, the
134  worst case $y(x) = x_1$ has all but one neuron unused. Therefore $\mathbb{E}\Delta = x_i \mathbb{E}\xi_i p$ and $\text{Var}\Delta = x_i^2 \text{Var}\xi_i = p(1-p) \sim p$,
135  variance does not decay with $n$.

136  This proposition gives sufficient conditions under which the weight perturbation is small and it is less and less
137  as $n$ increases and $p$ decreases:

138  **Proposition 3.** *Under A1,2 and AA1, for $\alpha > p$, the norm of the weight perturbation $U_l^i$ at layer $l$ is probabilistically*
139  *bounded as: $\delta_0 = \mathbb{P}\{\|U_l\|_1 \geq \alpha\|W_l\|\} \leq \exp\left(-n_l \cdot q \cdot d_{KL}(\alpha\|p_l)\right)$ with KL-divergence between numbers $a, b \in (0, 1)$,*
140  $d_{KL}(a, b) = a \log a/b + (1-a) \log(1-a)/(1-b)$

141  *Proof.*  See [5] for the proof details as this is a standard technique based on Chernoff bounds for Binomial distribution
142  with $p \to 0$.  These are a quantified version of the Law of Large Numbers (which states that an average of
143  identical and independent random variables tends to its mean). Specifically, Chernoff bounds show that the tail of
144  a distribution is exponentially small: $\mathbb{P}\{X \geq \mathbb{E}X + \varepsilon\mathbb{E}X\} \leq \exp(-c\varepsilon^2)$.

Specifically, if we consider a case $X = Bin(n, p)$ with $p \to 0$, for which $q = 1$, we have [5, 1] for $\alpha = k/n > p$:

$$\mathbb{P}[X \geq k] \leq \exp\left(-n D_{KL}\left(\frac{k}{n}\|\alpha\right)\right)$$

145  In case if we rewrite $k = \alpha n$, this gives us the result.
146  Specifically, if we consider $\|U_l\|_1 = \sum_i |U_l^i| \approx k|W_l|$.  Therefore, this probability is $\mathbb{P}[X \geq k] = P[\|U_l\|_1 \geq$
147  $\alpha\|W_l\|_1]$.  This shows that the probability that a fraction of Bernoulli successes is more and more concentrated
148  around its mean, $np$. Therefore, it is less and less probable that the this fraction is $\geq \alpha > p$.

Factor $q$ appears because in the analysis of the sum

$$\frac{\left(\sum_{i=1}^{n} |W^i|\right)^2}{\sum_{i=1}^{n} |W^i|^2} \geq \frac{n^2 W_{\min}^2}{n W_{\max}^2} = nq$$

149  $\qquad\qquad\qquad\qquad\qquad\qquad\qquad\qquad\qquad\qquad\qquad\qquad\qquad\qquad\qquad\qquad\qquad\qquad\qquad\qquad\qquad\qquad$ $\square$

## 150  4.2  Taylor Expansion for a Small Weight Perturbation

151  In this section, we develop a more precise expression for $\mathbb{E}\Delta$ and $\text{Var}\Delta$. Previously, we have seen that the perfect
152  fault-tolerant network has $\text{Var}\Delta = \mathcal{O}(p/n)$. In this section, we give sufficient conditions when complex neural
153  networks behave as the toy example from the previous section as well.

154  We would like to obtain a Taylor expansion of the expectation of random variable $\mathbb{E}\Delta = T_1 + T_2$ in terms of $p$
155  and $q = 1/n$ with $r = p + q$ where $T_1 = \mathcal{O}(p)$ and $T_2 = \mathcal{O}(r^2)$. For the variance, we want to have $\text{Var}\Delta = T_3 + T_4$
156  with $T_3 = \mathcal{O}(p/n)$ and $T_4 = \mathcal{O}(r^3)$. Our goal here is to make this expression decay as $n \to \infty$ as in the toy example.
157  We will show in Theorem 1, the first-order terms indeed behave as we expect. However, the expansion also contains
158  a remainder (terms $T_2$ and $T_4$). In order for the remainder to be small, we need additional assumptions. It is easy
159  to come up with an example for which the first term $T_1$ is zero, but the error is still non-zero, illustrating that the
160  remainder is important. Consider one neuron working and the rest having zero weights. Consider a summation of
161  outputs of the neurons, each with a quadratic activation. Then $y = (x_i - \alpha)^2$, and $\nabla_x y = 0$ at $x_i = \alpha$. In addition,
162  $\mathbb{E}\Delta = \Theta(p)$, however, the first term in Taylor expansion is $T_1 = 0$. The remainder here is $T_2 = O(p)$ and not $O(r^2)$,
163  no matter how many neurons we have. The problem with this example is discontinuity: one neuron with non-zero
164  weights is not at all like its neighbors. We thus show that discontinuity can lead to a lack of fault tolerance. Next,
165  we generalize this sketch and show that some form of continuity is sufficient for the network to be fault-tolerant.

First, we reiterate on the toy motivating example from the previous section. Consider a 1-layer neural network

$$y(x) = \sum_{i=1}^{n} w_i x_i$$

We assume that all neurons and inputs are used. Specifically, for the $q$-factor $q(x) = \max|x_i|/\min|x_i|$, we have $q(x) \approx q(w) \sim 1$. We are interested in how the network behaves as $n \to \infty$ (the infinite width limit). For the input, we want the magnitude of individual entries to stay constant in this limit. Thus, $|x_i| = \mathcal{O}(1)$. Now we look at the function $y(x)$ that the network computes. Since the number of terms grows, each of them must decay as $1/n$: $w_i \sim 1/n$. The simplest example has $x = (1, ..., 1)$ and $w = (1/n, ..., 1/n)$, which results in $y(x) \equiv 1$ for all $n$. Now we consider fault tolerance of such a network. We take $\Delta = \sum_{i=1}^{n} w_i((1 - \xi^i) - 1)x_i$ for $\xi^i \sim Be(p)$ being the indicator that the $i$'th input neuron has failed. Therefore, $\mathbb{E}\Delta = -p/n\sum x_i w_i = -p$, and $\text{Var}\Delta = \sum x_i^2 w_i^2 p(1-p) \sim p/n$. We see that the expectation does not change when $n$ grows, but variance decays as $1/n$. These are the values that we will try to obtain from real networks. Intuitively, we expect that the fault tolerance increases when width increases, because there are more neurons. This is the case for the simple example above. However, it is not the case if all but one neuron are unused. Then, the probability of failure is always $p$, no matter how many neurons there are. Thus, the variance does not decrease with $n$. We thus are interested in utilizing the neurons we have to their maximal capacity to increase the fault tolerance.

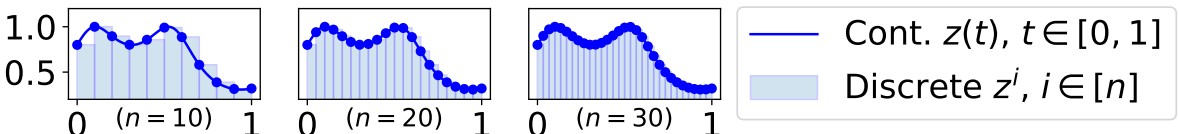

Figure 1: Discrete (standard) neural network approximating a continuous network

Overall, we have the following plan. We give an expansion of $\text{Var}\Delta$ in terms of $p$ and $q = 1/n$, and give sufficient conditions for which the remainder terms are small. We use the first term explicitly when regularizing the network. In order for the expansion to work, we formalize the difference between "all neurons doing the same task" and "all but one are unused". Specifically, we define a class of "good" networks for which fault tolerance is sufficient, via the *continuous limit* [23].

**Functions, functionals and operators.** We call maps from numbers to numbers as functions, maps from functions to numbers as functionals and maps from functions to functions as operators.

We consider a subset of the space of real-valued functions with domain $T$: $\mathcal{F}(T) = BPC^\infty(T)$ for $T = [0, 1]$. This is a space of bounded piecewise-continuous functions $f \in \mathcal{F}$, $f: T \to \mathbb{R}$ such that there is a finite set of points $\{x_i\} \subset T$ such that $f|_{(x_i, x_{i+1})} \in C^\infty(T)$ – infinitely-differentiable. We note that $\mathcal{F} \subset \mathcal{L}^1$ which means that $\int_{t \in T} |f(t)|dt < \infty$. In addition, $\int |f^k(t)| < \infty$. Moreover, for two $f, g \in \mathcal{F}$, $(f \cdot g)(t) = f(t) \cdot g(t) \in \mathcal{F}$.

**Additional Definition 1.** *(Continuous neural network) A continuous neural network [15] of function class $\mathcal{H}$ with $L$ layers is an operator $y_L : \mathcal{H} \to \mathcal{H}$ defined by a tuple $(L, W, B, \varphi)$ with a tuple of weights functions $W = (W_1, ..., W_L)$ with $W_l \in \mathcal{H}(T^2)$ (or their distributions), bias functions $B = (b_1, ..., b_L)$ (or their distributions) with $b_l \in \mathcal{F}$ by the expression $y_l(t) = \varphi(z_l(t))$, $z_l(t) = \int_{t' \in T} W_l(t, t')y_{l-1}(t')dt' + b_l(t)$, $l \in \overline{1, L}$, $z_0 = y_0 = x$ and $y_L(t) = z_L(t)$.*

We note that a regular neural networks has $T_l = [n_l]$. Continuous networks were re-introduced in [15]. The classes of discrete and continuous networks can be generalized as Deep Function Machines [12], as a special case when $L_1$-norms of inputs and activations of discrete networks approximating continuous ones increase linearly with $n$. In [23], depth of a network is also considered to be continuous. Continuous networks are also called the integral representation [23]. Previous work considers other limits rather than $L_1$-norm growing linearly, but here we only consider this case.

Next, we connect continuous and regular (discrete) networks. To do that, we define how each quantity transforms when $n$ grows.

**Additional Definition 2.** *(Continuous-discrete correspondence with $\|x\|_1 = \mathcal{O}(n_0)$) Consider a continuous $NN_c = (L, W_c, B_c, \varphi)$ and a discrete one $NN_d = (L, W_d, B_d, \varphi)$. We consider a distribution of continuous inputs $\mathcal{F} \ni x \sim P_c$ and a distribution of discrete inputs $\mathbb{R}^n \ni x \sim P_d$. These distributions are linked: each element from $P_c$ corresponds to exactly one from $P_d$. We define the approximation error $A$ as*

$$A = \sup_{(x_c, x_d) \sim P_c \times P_d} \left[ \sup_{0 \leq l \leq L} \left[ \sup_{1 \leq i \leq n_l} \left| (y_l^d)^i - (y_l^c)\left(\frac{i-1}{n_l-1}\right) \right| \right] \right]$$

206 We define $n = \min\{n_0, ..., n_L\}$ the minimal width of the discrete network. If a series of discrete networks $NN_n$
207 has $A_n \to 0$, $n \to \infty$ for some $NN_c$, we say that $NN_n \to NN_c$.

208 The summary of correspondences between discrete and continuous quantities is shown in Table 1

| Quantity | Discrete (D1) | Continuous (AD1) | Relation (AD2, AP4) |
|---|---|---|---|
| Input | $x : [n_0] \to \mathbb{R}$ | $x : [0,1] \to \mathbb{R}$ | $x_i \approx x\left(\frac{i-1}{n_0-1}\right)$ |
| Norm | $\|x_d\|_1 = \sum_i |x_i|$ | $\|x_c\|_1 = \int |x(t)|dt$ | $\frac{1}{n_0}\|x_d\|_1 \approx \|x_c\|_1$ |
| Bias | $b_l : [n_l] \to \mathbb{R}$ | $b_l : [0,1] \to \mathbb{R}$ | $b_l^i \approx b_l\left(\frac{i-1}{n_l-1}\right)$ |
| Weights | $W_l : [n_l] \times [n_{l-1}] \to \mathbb{R}$ | $W_l : [0,1]^2 \to \mathbb{R}$ | $W_l^{ij} \approx \frac{1}{\mathbf{n_{l-1}}}W_l\left(\frac{i-1}{n_l-1}, \frac{j-1}{n_{l-1}-1}\right)$ |
| Pre-activations | $z_l^i = \sum_j W_l^{ij} y_{l-1}^j + b_l^i$ | $z_l(t) = \int_0^1 W_l(t, t')y_{l-1}(t')dt' + b_l(t)$ | $z_l^i \approx z_l\left(\frac{i-1}{n_l-1}\right)$ |
| Number of changes | $C_d = \sum_{ij}\left|W_l^{i+1,j} - W_l^{ij}\right|$ | $C_c = \int\int dtdt'|W_t'(t,t')|$ | $C_d \approx C_c$ |

Table 1: Correspondence between discrete and continuous quantities

209 **Additional Proposition 3.** *Continuous networks with $\mathcal{H} = \mathcal{F}$ are universal approximators*

210 *Proof.* Based on the proof of [15]. By the property of discrete networks, they are universal approximators [15].
211 Take a discrete network $y$ with a sufficiently low error, and define a continuous network $NN_c$ by using a piecewise-
212 constant $W$ and $b$ from $y$. Then, $NN_c \equiv NN_d$. The weights and biases are bounded and piecewise-continuous with
213 $n_l$ discontinuities at each layer. In addition, all functions are bounded since the weights are finite. □

214 In the following we will always use $\mathcal{H} = \mathcal{F}$, as it is expressive enough (AP3), and it is useful for us. We give a
215 sufficient condition for which $A_n \to 0$ as $n \to \infty$.

**Additional Proposition 4.** *If a discrete network $NN_n$ is defined from a continuous network $NN_c$ in the following
way, then $NN_n \to NN_c$ with error $A = \mathcal{O}(1/n)$. See (Figure 1)*

$$x^i = x\left(\frac{i-1}{n_0-1}\right), \ W_l^{ij} = \frac{1}{n_{l-1}}W_l\left(\frac{i-1}{n_l-1}, \frac{j-1}{n_{l-1}-1}\right), \ b_l^i = b_l\left(\frac{i-1}{n_l-1}\right)$$

216 In the following we write $\hat{i}_l = \frac{i-1}{n_l-1}$ for an index $i = 1..n_l$, as the range for each of the indices is known.

217 *Proof.* For layer 0, the error is 0 by definition of $x^i$. Also, $x$ and its derivatives are bounded. Suppose we
218 have shown that the error for layers $1..l-1$ is $\leq \varepsilon$, and that $y_{l'}$ is bounded with globally bounded deriva-
219 tives. Consider $\left|(y_l^d)^i - (y_l^c)\left(\frac{i-1}{n_l-1}\right)\right| = |b_l^i - b_l^i + \sum_j W_{ij}y_{l-1}^j - \int W(\hat{i}, t')y_{l-1}(t')dt'| \leq |1/n_{l-1}\sum_j W(\hat{i}, \hat{j})(y_{l-1}^j +$
220 $\varepsilon) - \int W(\hat{i}, t')y_{l-1}(t')dt'| \leq \varepsilon \int |W|dtdt' + |\int f(t)dt - 1/n \sum f(\hat{j})|$ for $f(t) = W(\hat{i}, t)y_{l-1}(t)$. The second term
221 is a Riemann sum remainder and can be upper-bounded as $\frac{|T|^2}{2n}\sup|f'(t)|$. We note that $|[0,1]| = 1$ and that
222 $|f'(t)| = |W_t'(\hat{i}, t)y_{l-1}(t) + W(\hat{i}, t)y_{l-1}'(t)| \leq \sup|W'|\sup|y| + \sup|W|\sup|y'| < \infty$ and does not depend on $n$ or the
223 input, as the bound is global. Therefore, the error at layer $l$ is $\leq (\varepsilon + 1/n)C \leq D/n$ for the initial choice $\varepsilon = 1/n$.
224 Thus, the full error is decaying as $1/n$: $A_n = \mathcal{O}(1/n)$. □

225 Using that result, we conclude that for any sufficiently smooth operator $y$, there exist a sequence $NN_n$ ap-
226 proximating $y$. First, by [15], continuous networks are universal approximators. Taking one and creating discrete
227 networks from it (as in AP4) gives the desired result as $\sup_x \|y_c - y_d\|_\infty \leq A = \mathcal{O}(1/n) \leq \varepsilon$ and $\|y_c - y\| \leq \varepsilon$ by the
228 approximation theorem for continuous nets.

229 **Assumption 3.** *We assume that $NN_n \to NN_d$. We assume that there are global reasonable bounds on the
230 derivatives of the function we want to approximate*

$$\sup_{x \sim P}\left|\frac{\delta^k y_L(t_L)}{\delta y_l(i_l^1)...\delta y_l(i_l^k)}\frac{1}{(i_l^1, ..., i_l^k)!}\right| \leq D_k, \ D_{a:b} = \max\{D_a, D_{a+1}, ..., D_b\}$$

231 Here for $i_s \in [n]$ and $q_j = |\{i_s = j\}|$ we define $(i_0, ..., i_k)! = \frac{n!}{q_1!...q_n!}$ a multinomial coefficient. In the paper, we
232 only need $k \in \{1, 2\}$.

The limit means that close-by neurons inside a discrete network compute similar functions. This helps fault tolerance because neurons become redundant. There can be many of such sequences $NN_n$, but for the sake of clarity, one might consider that gradient descent always converges to the "same way" to represent a function, regardless of how many neurons the network has. This can be made more realistic if we instead consider the distribution of continuous networks to which the gradient descent converges to. In this limit, the distribution of activations at each layer (including inputs and outputs) stays the same as $n$ grows, and only the number of nodes changes. Note that this limit makes neurons ordered: their order is not random anymore. The limit works when $n_l$ are sufficiently large. After that threshold, the network stops learning new features and only refines the ones that it has created already.

The derivative bound part of A3 can be enforced if there is a very good fit with the ground truth function $\tilde{y}(x)$. Indeed, if $\tilde{y}(x) \equiv y_L(x)$, then the derivative of the network depends on the function being approximated. For discrete-continuous quantities we write $X_d \approx X_c$ meaning that $|X_d - X_c| \le \varepsilon$ for a sufficiently large $n$.

We note that our new assumptions extend the previous ones:

**Additional Proposition 5.** *A3 results in AA1 with*

$$q = \frac{\left(\int \int |W(t,t')| dt dt'\right)^2}{\int \left(\int W(t,t')| dt\right)^2 dt'} + o(1),\ n_l \to \infty$$

*Proof.* A simple calculation:

$$\sum |W^i| = n \cdot \frac{1}{n} \sum \sum_{ij} |W^{ij}| \to n \int |W(t,t')| dt dt', \ \sum |W^i|^2 \to n \int \left(\int |W(t,t') dt 1|\right)^2 dt$$

Dividing these two gives the result. □

**Can we make the input norm bounded?** Sometimes, input data vectors are normalized: $\|x\| = 1$ [14]. In our analysis, it is better to do the opposite: $\|x\|_1 = \mathcal{O}(n_0)$. First, we want to preserve the magnitude of outputs to fit the same function as $n$ increases. In addition, the magnitude of pre-activations must stay in the same range to prevent vanishing or exploding gradients. We extend that condition to the input layer as well for convenience. Another approach would be to, for example, keep the norm $x$ constant as $n$ grows. In that case, to guarantee the same output magnitude, input weights must be larger. This is just a choice of scaling for the convenience of notation. We note that inputs can be still normalized *component-wise*.

**Less weight decay leads to no fault tolerance.** Consider the simplest case with Gaussian weights: $y = w^T x$, $w_i \sim N(0, \sigma^2)$. Then $y \sim N(0, \sigma^2 \|x\|_2^2)$. In order to avoid vanishing gradients, we set $\sigma^2 \|x\|_2^2 = const$. If $\|x\|_2^2 = \mathcal{O}(n)$, then we must set $\sigma^2 \sim 1/n$ for the variance of the output to stay constant. This is a well-known initialization technique preserving the *variance* of pre-activations and thus eliminating vanishing or exploding gradients. In contrast to this, by A3 we preserve the *mean* activation at each neuron instead, like for stochastic neural networks [24]. In the Gaussian case, the error $\Delta = -\sum \xi^i x_i w_i$. $\mathbb{E}\Delta = 0$ and $\text{Var}\Delta = \sum x_i^2 \mathbb{E}(\xi^i)^2 \mathbb{E}w_i^2 = \sigma^2 p \|x\|_2^2 = p \cdot const$. This does not decay with $n$. A more formal statement can be found in AP6.

**The NTK limit.** In the NTK limit [14], there is a decay of the variance $\text{Var}\Delta$ since both $\|x\|_2 = 1$ and $\sigma^2 \sim 1/n$, but that happens because variance decreases in any case, not related to fault tolerance, since $\sigma^2 \|x\|_2^2 \sim 1/n$. The "lazy" regime [7] of the NTK limit implies that every hidden neuron is close to its initialization, which is random. Thus, there is no continuity in the network: close-by neurons do not compute similar functions. Thus, the NTK limit is incompatible with our continuous limit.

Below we formalize the claim that a network with constant pre-activations variance is not fault-tolerant. This means that an untrained network with standard initialization is not fault tolerant.

**Additional Proposition 6.** *The fault tolerance error variance for a network $y = w^T x$ with $\mathbb{E}w_i = 0$ and $\text{Var}[y(x)] = const$ does not decay with $n$*

*Proof.* Consider $y(x) = \sum w_i x_i$ and $\Delta = -\sum w_i x_i \xi^i$. Since $\mathbb{E}w_i = 0$, $\mathbb{E}y(x) = \mathbb{E}\Delta = 0$. In addition, $\text{Var}y(x) = \sum \mathbb{E}w_i^2 \mathbb{E}(x^i)^2 = const$. Then, $\text{Var}\Delta = \sum \mathbb{E}w_i^2 \mathbb{E}(x^i)^2 \mathbb{E}(\xi^i)^2 = p\text{Var}y(x) = p \cdot const$, not decaying with $n$ □

Next, we want to harness the benefits of having a continuous limit by A3. We want to bound the Taylor remainder, and for that, we bound higher-order derivatives. We bound them via the operator derivative of the continuous limit.

**Operator derivatives.** The derivatives for functionals and operators are more cumbersome to handle because of the many arguments that they have (the function the operator is taken at, the function the resulting derivative operator acts on, and the arguments for these functions), so we explicitly define all of them below. We consider operators of the following form, where $x \in \mathcal{F}$ and $Y[x] \in \mathcal{F}$ are functions:

$$Y[x](t) = \sigma\left(\int W(t, t')x(t')dt'\right)$$

This is one layer from AD1. We define the *operator derivative* of $Y[x]$ point-wise with the RHS being a *functional derivative*:

$$\left(\frac{\delta Y[x]}{\delta x}\right)(t) = \frac{\delta(Y[x](t))}{\delta x}$$

For functionals $F[x]$ we use the standard functional derivative:

$$\frac{\delta F[x]}{\delta x}[\varphi] = \lim_{\varepsilon \to 0} \frac{F[x + \varepsilon\varphi] - F[x]}{\varepsilon}$$

For our case $F[x] = F_t[x] = Y[x](t)$, we have

$$\frac{\delta F[x]}{\delta x}[\varphi] = \lim_{\varepsilon \to 0} \varepsilon^{-1} \cdot \left(\sigma\left(\int W(t, t')(x(t') + \varepsilon\varphi(t'))dt'\right) - \sigma\left(\int W(t, t')x(t')dt'\right)\right) = \sigma'(F[x])\int W(t, t')\varphi(t')dt'$$

Next, we consider the functional derivative *at a point*. This quantity is similar to just one *component* of the gradient of an ordinary function, with the complete gradient being similar to the functional derivative defined above. We define the derivative at a point $\delta F[x]/\delta x(s)$ via the Euler-Lagrange equation, since we only consider functionals of the form $F_t[x] = Y[x](t)$ for some fixed $t$ which are then given by an integral expression

$$F_t[x] = Y[x](t) = \sigma\left(\int \underbrace{W(t, t')x(t')}_{L(t', x(t'), x'(t'))} dt'\right)$$

We define the inner part $G[x] = \int W(t, t')x(t')dt'$, thus, $F[x] = \sigma(G[x])$. In this case, since the integral only depends on the function $x$ explicitly, but not on its derivatives, the functional derivative at point $s$ is

$$\frac{\delta F[x]}{\delta x(s)} = \sigma'(G[x])\left(\frac{\partial L}{\partial x} - \frac{d}{ds}\frac{\partial L}{\partial x'}\right) = \sigma'(G[x])\frac{\partial L}{\partial x} = \sigma'(G[x])W(t, s)$$

The definition of a functional derivative at a point $\delta F[x]/\delta x(s)$ ("component of the gradient") can be reconciled with the definition of the functional derivative $\delta F[x]/\delta x$ ("full gradient") if we consider the Dirac delta-function:

$$\frac{\delta F[x]}{\delta x}[\varphi = \delta(t' - s)] = \sigma'(G[x])\int W(t, t')\delta(t' - s) = \sigma'(G[x])W(t, s) \equiv \frac{\delta F[x]}{\delta x(s)}$$

We define the *operator derivative at a point* in a point-wise manner via the functional derivative at a point:

$$\left(\frac{\delta Y[x]}{\delta x(s)}\right)(t) = \frac{\delta Y[x](t)}{\delta x(s)}$$

Having that definition, we compute for our case $(\delta Y[x]/\delta x(s))(t) = \sigma'(G_t[x])W(t, s)$.

Now we see that the rules for differentiating operators *in our case* are the same as the well-known rules for the derivatives of standard vector-functions. Indeed, if we consider $y_i(x) = \sigma(\sum_j W_{ij}x_j)$ with the inner part $g_i(x) = \sum_j W_{ij}x_j$ giving $y_i(x) = \sigma(g_i(x))$, then $\partial y_i/\partial x_k = \sigma'(g(x))W_{ik}$. This looks exactly like the expression for $\partial Y[x](t)/\partial x(s)$. By induction, this correspondence holds for higher derivatives as well. However, this does not imply that these quantities are equal. In fact, we will show that they differ by a factor of $1/n_l^k$ where $k$ is the order of the derivative.

We characterize the derivatives of a discrete NN $\partial^k y_L/\partial y_l^{i^1}...\partial y_l^{i^k}$ in terms of operator derivatives of a continuous NN $\delta^k y_L/\delta y_l(i_l^1)...\delta y_l(i_l^k)$. We only assume the continuous limit:

**Proposition 4.** *For a sequence $NN_n \to NN_c$ with $\varphi \in C^\infty$, the derivatives decay as:*

$$\frac{\partial^k y_L}{\partial y_l^{i_l^1}...\partial y_l^{i_l^k}} = \frac{1}{n_0^k}\frac{\delta^k y_L}{\delta y_l(i_l^1)...\delta y_l(i_l^k)} + o(1),\ n_l \to \infty$$

Intuitively, this means that the more neurons we have, the less is each of them important. First, consider a simple example $y = \sigma(\sum w_i x_i)$. Here the weight function is $w_i = 1/n$, $w(\hat{i}) = 1$, $x_i = 1$, and $A = 0$. Then $\partial y / \partial x_i = \sigma'(\cdot) w_i \sim 1/n$ and $\partial^2 y / \partial x_i \partial x_j = \sigma''(\cdot) w_i w_j \sim 1/n^2$. We note that the expression inside the sigmoid has a limit and it's close to the integral by continuity of $\sigma'$ and $\sigma''$.

*Proof.* Now we prove P4. Consider the first and second derivatives. Note that the operator derivatives only depend on the number of layers and the dataset. It does not depend on any $n$ anymore.

$$
\begin{aligned}
\frac{\partial y_L^{i_L}}{\partial x_{i_0}} &= \sum_{i_{L-1},\ldots,i_1} W_L^{i_L i_{L-1}} \ldots W_1^{i_1 i_0} \sigma'(z_L^{i_L}) \ldots \sigma'(z_1^{i_1}) && \text{By definition of a neural net} \\
&\approx \frac{1}{n_0} \int dt_1 W_1(t_1, t_0) \sigma'(z_1(t_1)) \ldots \int dt_{L-1} W_{L-1}(t_{L-1}, t_{L-2}) \cdot W_L(t_L, t_{L-1}) \cdot \sigma'(z_{L-1}(t_{L-1})) && \text{continuous limit} \\
&= \frac{1}{n_0} \frac{\partial y_L(i_L)}{\partial x(i_0)} && \text{operator} \sim \text{ordinary}
\end{aligned}
$$

Crucially, the factor $1/n_0$ appears because we **do not sum** over $i_0$, as it is fixed, but we have a weight vector $W_1 \sim 1/n_0$ nevertheless. For all other indices, we have a weight matrix $W_l \sim 1/n_{l-1}$ **as well as** a summation over $i_{l-1}$.

$$
\begin{aligned}
\frac{\partial^2 y_L^{i_L}}{\partial x_{i_0} \partial x_{i_0'}} &= \sum_{i_{L-1} \ldots i_1} W_L^{i_L i_{L-1}} \ldots W_1^{i_1 i_0} \sum_{s=1}^{L} \sigma''(z_s^{i_s}) \prod_{s' \neq s} \sigma'(z_{s'}^{i_{s'}}) \sum_{i_s' \ldots i_1'} W_s^{i_s' i_{s-1}'} \ldots W_1^{i_1' i_0'} && \text{From previous} \\
&\approx \frac{1}{n_0^2} \frac{\delta^2 y_L(i_L)}{\delta x(i_0) \delta x(i_0')} && \text{continuous+operator} \sim \text{ordinary}
\end{aligned}
$$

Here, the factor $1/n_0^2$ appears because we never sum over $i_0$, but the weight matrix $W_1 \sim 1/n_0$ appears twice. $\square$

**When does the limit hold?** Now, we have *assumed* that a network has a continuous limit in A3. However, this might not be the case: the NTK limit [14] is an example of that, as weights there stay close to their random initialization [7], thus, they are extremely dissimilar.

1. **Explicit duplication.** We copy each neuron multiple times and reduce the outgoing weights. If we set $W(t, t')$ to be piecewise-constant, then the approximation error is zero $A = 0$, and it does not depend on the degree of duplication. This is the obvious case where the network is becoming more fault-tolerant using duplication, and our framework confirms that. The problem with explicit duplication of non-regularized networks is that their fault-tolerance is suboptimal. Not all neurons are equally important to duplicate. Thus, it's more efficient to utilize all the neurons in the best way for fault tolerance by duplicating only the important ones.

2. **Explicit regularization.** We make adjacent neurons compute similar functions, thus, allowing for redundancy. We first consider the local "number of changes" metric (Table 1). Specifically, for some function $z(t) \in [0, 1]$, $\int |z'(t)| dt$ represents how many times the function goes fully from 0 to 1 or vice versa. Our idea is to use that for the weights to quantify their discontinuity:

$$
\begin{aligned}
C_1 = \sum_{ij} |W_l^{ij} - W_l^{i+1,j}| &= \frac{1}{n_0} \sum_{ij} |W(\hat{i}, \hat{j}) - W(i \hat{+} 1, \hat{j})| \\
&= \frac{1}{n_0} \sum_{ij} \left| W\left(\frac{i-1}{n_1-1}, \frac{j-1}{n_0-1}\right) - W\left(\frac{i}{n_1-1}, \frac{j-1}{n_0-1}\right) \right| \\
&\approx \frac{1}{n_0 n_1} \sum_{ij} \left| W_t'\left(\frac{i-1}{n_1-1}, \frac{j-1}{n_0-1}\right) \right| \\
&\approx \int |W_t'(t, t')| dt dt'
\end{aligned}
$$

The term above, if small, guarantees that, for each *input* neuron, neighboring output neurons will use it in similar ways. The same is applied for $W^T$ as well: $C_2 = n_0/n_1 \sum_{ij} |W^{ij} - W^{i,j+1}| \approx \int |W'_{t'}(t,t')| dt dt'$ which guarantees that, for each *output* neuron, neighboring input neurons are used similarly by it.

In addition to making adjacent neurons computing similar functions, we add another term $C_3$ by Gaussian smoothing: a Gaussian kernel is convolved with the weights, and the weights are subtracted from the result. The difference shows how much the current value differs from an aggregated local "mean" value.

We explicitly enforce the continuous limit by adding a regularization term of $smooth(W) := C_1 + C_2 + C_3$. Here *smooth* consists of three parts, see `FilterPlayground.ipynb`. The first part $C_1, C_2$ computes the numerical derivative with $2, 4, 10, 14$ points in the array. The second part convolves the input with a Gaussian kernel and subtracts the original, resulting in a measure of discontinuity. All metrics are normalized to work with any size, so that scaling the network up does not change the magnitude of $C_i$ significantly. The implementation can be found in `continuity.py`.

We check the derivative decay prediction (P4) which must follow from the continuity assumption A3, experimentally. We run experiments for the MNIST dataset and architecture $(784, N, 100, 10)$ – 2 hidden layers with sigmoid activations, unit Lipshitz coefficient, batch size of 1000, and $N$ from 50 to a few thousand. We measure $\|W\|_\infty = \max_i \sum_j |W^{ij}| \approx \max_t \int_{t'} |W(t,t')| dt'$ which should stay constant (we measure the product of these over layers $l$, as only the total norm is expected to stay constant, whereas individual layers can change the magnitude of $W$). We also measure the components of the first derivative $avg(|D_i|)$ and the components of the Hessian $avg_i(|H_{ii}|)$, $avg_{(ij)}(|H_{ij}|)$ which we expect to decay with $n$ as by P4. We show that without regularization, the weights $\|W\|_\infty$ increase, see `WeightDecay-FC-MNIST.ipynb`, and the derivatives and Hessians do decay but with a smaller rate, insufficient for T1 to work, see `DerivativeDecay-FC-MNIST.ipynb`. We repeat each experiment 10 times and report mean and standard deviation, see Figure 2. In contrast, our proposed regularization results in a smooth transition between neurons at each layer. They are grouped by their similarity. This can be visually seen in `WeightDecay-Continuity-FC-MNIST.ipynb` or in Figure 2. Contrary to networks without regularization, first layer weight profiles seem to have a meaningful image, and these images are similar for close-by neurons. There is also an increase in continuity of activations for a fixed input. For this regularization technique, we show that the product of the weights stabilizes, and the derivatives decay with proper slopes of $-1$ and $-2$. The accuracy, however, drops from 98% to 90%. We note that this does not demonstrate that our approach necessarily leads to a decrease in accuracy, as continuous networks are general by AP3.

We test the result on Fashion MNIST as well (Figure 3, notebook names are the same but with `Fashion`) and report similar behavior. On the Boston Housing dataset, non-regularized networks already have the desired characteristics (Figure 4).

> **Note:** the condition above of $C_1 + C_2 + C_3$ being small is, strictly speaking, only a *necessary* condition for A3, but not a sufficient one. Even if networks are smooth enough, they might not have a limit $NN_n \to NN_c$, as they could implement the function in drastically different ways: for example, the networks $NN_n$ can be all smooth, but approximate *different* continuous functions. However, this condition *is* sufficient for the derivatives $D_k$ to stay constant (second part of A3), which is the only requirement for T1 to work. [a] Thus, our approach can give a formal guarantee of fault tolerance. An attempt to give a *truly* sufficient condition for A3 would be to train the bigger network given duplicated weights of a smaller network, and penalizing a bigger network from having weights different from their initialization.
>
> **Different smoothing techniques.** Currently, $C_2$ is unused in our implementation, as it was sufficient to use $C_1 + C_3$ to achieve the fault tolerance required in our experiments. Another method to make close-by neurons similar could be to use some clustering loss in the space of weights accounting for locality, like the Kohonen self-organizing map. One more idea is to regularize the weights with a likelihood of a Gaussian process, enforcing smoothness and a required range of dependencies. Another idea is to use the second derivative, which is connected to the curvature of a curve $(x, y(x))$ in the 2D space: $\kappa = y''/(1 + (y')^2)^{3/2}$. The interpretation here is that $\kappa = 1/R$ for $R$ being the radius of curvature, a geometrical property of a curve showing how much it deviates from being a line.

---

[a] A discussion of why $D_k$ stay constant is given in the analysis of the correctness for Algorithm 1.

3. **Convolutional networks.** For images, this limit naturally corresponds to scaling the images [12] with intermediary pixels being just added between the original ones. Images are piecewise-continuous because

adjacent pixels are likely to belong to the same object having the same color, and there are only finitely many objects on the image. Convolutional networks, due to their locality, fit our assumption on one condition. We need to have a large enough kernel size, as otherwise the non-smoothness is high. Specifically, for CNNs, $C_1$ is small, as neighboring neurons have similar receptive fields. In contrast, $C_2$ can be large in case if the kernel size is small: for example, the kernel $(-1, 1)$ in the 1D case will always result in a high discontinuity: the coefficients are vastly different and require more values in between them to allow for a continuous limit and redundancy.

The notebook `ConvNetTest-VGG16-ManyImages.ipynb` investigates into this. We note that we do not present this result in the main paper and make it a future research direction instead. In this paper, we give qualitative description of applicability of our techniques to convolutional networks (big kernel size for a small $C_2$ with respect to kernel size, smooth activation and pooling to allow for T1 to work), as it would be out of the scope of a single paper to go into details.

4. **Theoretical considerations for the general case.**

How does the network behave when the number of neurons increases, and it is trained with gradient descent from scratch? First, there are permutations of neurons, which we ignore. Secondly, there could be many ways to represent the same function. One constraint is that the magnitude of outputs in the output layer is preserved. Intermediate layers need to have a non-vanishing pre-activation values to overcome vanishing/exploding gradients. In addition, input limit might be enforced such that $x_i \approx x(\hat{i})$. Now, gradient descent results in a discrete network $NN_d$ which can be seen as a discretization of some continuous network $NN_c$. Since NN's derivatives are globally bounded, GD converges to a critical point. Each critical point determines the range of initializations which lead to it, partitioning the whole space into regions. Each fixed point with a sufficiently low loss thus corresponds to a set of continuous networks "passing through" a resulting discrete network. Each of the continuous networks can have different implementations in discrete networks of larger size. We choose a path in networks of different sizes $n$ and denote the probability $s_n$ over initializations to choose that particular continuous network. Therefore, on that path, derivatives decay as we want since $NN_n \to NN_c$. The problem might arise if a particular continuous limit $NN_c$ has an extremely small probability (over initializations) of gradient descent giving $NN_n$: if $s_n \to 0$, this particular network $NN_c$ is unlikely to appear. We leave the study of this as a future research direction.

Now we use the derivative decay from P4 to show fault tolerance using a Taylor expansion. We write $q = 1/n_l$ and $r = p + q$. In the following we will use Assumption 3 only by its consequence – Proposition 4. We note that the conclusion of it can hold in other cases as well. We just give sufficient conditions for which it holds.

**Theorem 1.** *For crashes at layer $l$ and output of layer $L$ under assumption 3 the mean and variance of the error can be approximated as*

$$\mathbb{E}\Delta_L = p_l \sum_{i=1}^{n_l} \frac{\partial y_L}{\partial \xi_l^i} + \Theta_{\pm}(1)D_2 r^2, \ \mathrm{Var}\Delta_L = p_l \sum_{i=1}^{n_l} \left(\frac{\partial y_L}{\partial \xi_l^i}\right)^2 + \Theta_{\pm}(1)D_{12}^2 r^3$$

*By $\Theta_{\pm}(1)$ we denote any function taking values in $[-1, 1]$. The derivative $\partial y_L / \partial \xi_l^i(\xi) \equiv -\partial y_L(y_l - \xi \odot y_l)/\partial y_l^i \cdot y_l^i$ is interpreted as if $\xi_l^i$ was a real variable.*

*Proof.* We consider crashes at layer $l$ as crashes in the input $x$ to the rest of the layers of the network. Thus, without loss of generality, we set $l = 0$.

Consider $\Delta(\xi) = y(\underbrace{(1 - \xi) \odot x}_{\hat{x}(\xi)}) - y(x)$. Then we explicitly compute

$$\frac{\partial \Delta(\xi)}{\partial \xi^i} = -\frac{\partial y(\hat{x}(\xi))}{\partial x_i} x_i, \ \frac{\partial^2 \Delta(\xi)}{\partial \xi^i \partial \xi^j} = \frac{\partial^2 y(\hat{x}(\xi))}{\partial x_i \partial x_j} x_i x_j$$

Now, consider $\Delta(\xi) = \Delta(0) + (\Delta'(0), \xi) + \frac{1}{2}\xi^T \Delta''(t(\xi))\xi$ by the Taylor theorem with a Lagrange remainder.

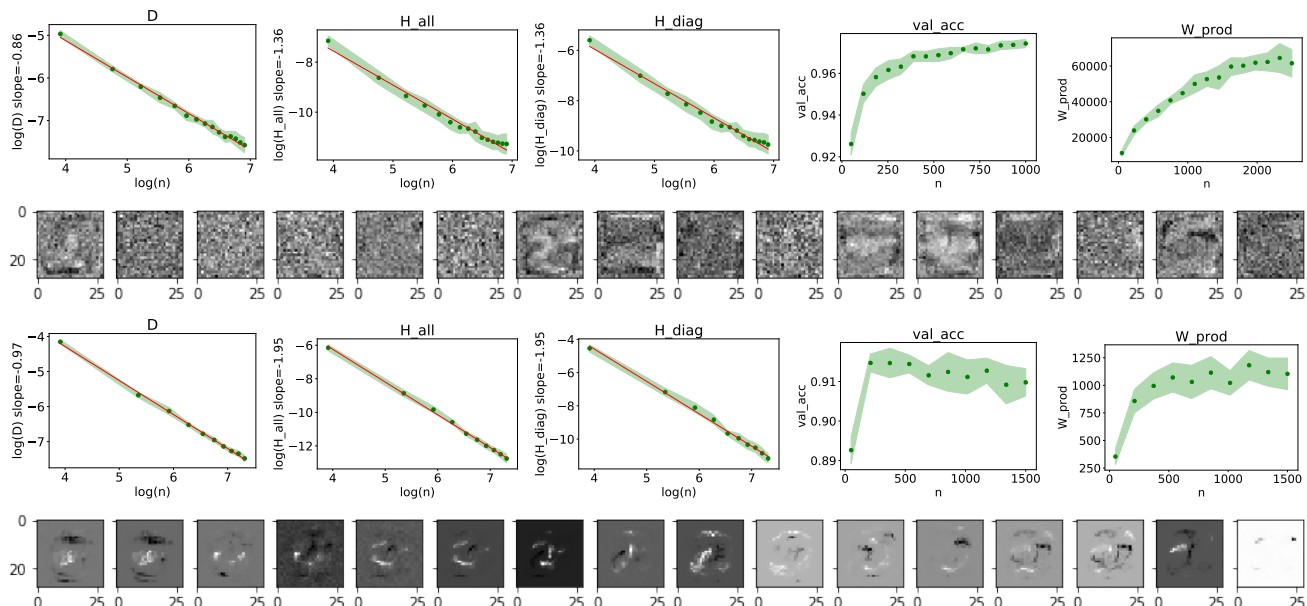

Figure 2: Non-regularized MNIST (first two rows), regularized MNIST (second two rows). Odd rows show the charts of the decay of first derivatives (D) and second derivatives (average over all $i, j$ Hessian's components $|H_{ij}|$, and only the diagonal elements $|H_{ii}|$). Next, validation accuracy is shown as well is the product of the infinity norms of the weights $\|W_L\|_\infty \cdot ... \cdot \|W_1\|_\infty$. Even rows show first-layer weights as images

We assume $\|x\|_\infty \leq 1$ (otherwise we rescale $W_1$). We group the terms into distinct cases $i = j$ and $i \neq j$:

$$\mathbb{E}\Delta = -p\Delta'(0) + \underbrace{\sum_{i=j} \mathbb{E}(\xi^i)^2 \Delta''(t(\xi))}_{E_1} + \underbrace{\sum_{i \neq j} \mathbb{E}\xi^i \Delta''(t(\xi))\xi^j}_{E_2}$$

The second term is $|E_1| \leq p \cdot 2D_2/n_0^2 \cdot n_0 = \mathcal{O}(pD_2/n_0) = (pqD_2)$

The third term is $|E_3| \leq n_0^2 D_2 p^2/n_0^2 = \mathcal{O}(p^2 D_2)$.

Therefore, we have an expansion $\boxed{\mathbb{E}\Delta = -\nabla_\xi \Delta(0) \cdot x + (q+p) \cdot pD_2 = -p \sum_i x_i \frac{\partial y(x)}{\partial x_i} + \mathcal{O}(D_1 p + D_2 r^2)}$.

The expectation just decays with $p$, but not with $n_0$.

Now, consider the variance $\text{Var}\Delta = \text{Var}(\underbrace{\Delta'(0)\xi}_{V_1} + \underbrace{0.5\xi^T \Delta''(t(\xi))\xi}_{V_2}) = \mathbb{E}\Delta^2 - (\mathbb{E}\Delta)^2 = \mathbb{E}(V_1 + V_2)^2 - (\mathbb{E}(V_1 + V_2))^2 =$

$\text{Var}V_1 + \text{Var}V_2 + 2Cov(V1, V2)$

Consider $\text{Var}V_1 = p \sum (\frac{\partial y(x)}{\partial x_i})^2 x_i^2 = \mathcal{O}(pqD_1)$. This is the leading term, the rest are smaller.

And the second term $\text{Var}V_2 \leq \mathbb{E}V_2^2 = \sum_{ijkl} D_2^2/n_0^4 \mathbb{E}\xi^i \xi^j \xi_k \xi_l$. Here we consider various cases for indices $i, j, k, l$, based on the partition of $4 = 0 + 4 = 2 + 2 = 1 + 3$. If all indices are different, we get $\mathcal{O}(p^4)$ from $\mathbb{E}\xi$. If all are the same, we get $\mathcal{O}(p/n^3)$. If we have 2 groups of 2, we get $\mathcal{O}(p^2/n^2)$, if $3+1$ we get $\mathcal{O}(p^2/n^2)$. Thus, $\text{Var}V_2 = \mathcal{O}(D_2^2 r^4)$

Consider the final term $Cov(V_1, V_2) \leq \mathbb{E}|V_1 V_2| + \mathbb{E}|V_1|\mathbb{E}|V_2|$. $\mathbb{E}|V_1| \leq \mathcal{O}(pD_1)$, $\mathbb{E}|V_2| \leq D_2 p/n$. $\mathbb{E}V_1 V_2 = \sum_{ijk} D_1 D_2/n_0^3 \mathbb{E}\xi^i \xi^j \xi_k$. All different indices give $p^3$, all same indices give $p/n_0^2$, and $2+1$ give $p^2/n$. Thus, $Cov(V_1, V_2) = \mathcal{O}(D_1 D_2 r^3)$

Finally, $\boxed{\text{Var}\Delta = p\nabla^2 \Delta(0)x + \mathcal{O}(r^3 D_1 D_2) + \mathcal{O}(r^4 D_2^2) = \sum \left(\frac{\partial y(x)}{\partial x_i}\right)^2 x_i^2 + \mathcal{O}(r^3 D^2) = \mathcal{O}(D_1^2 p/n + D^2 r^3)}$ $\square$

**A better remainder.** It is possible to obtain a remainder of $\mathcal{O}(p^2/n) + \mathcal{O}(p/n^2)$ for the variance instead of a generic $\mathcal{O}(r^3)$. This makes the remainder decay with $n$ as well. A more fine-grained analysis of the remainder could

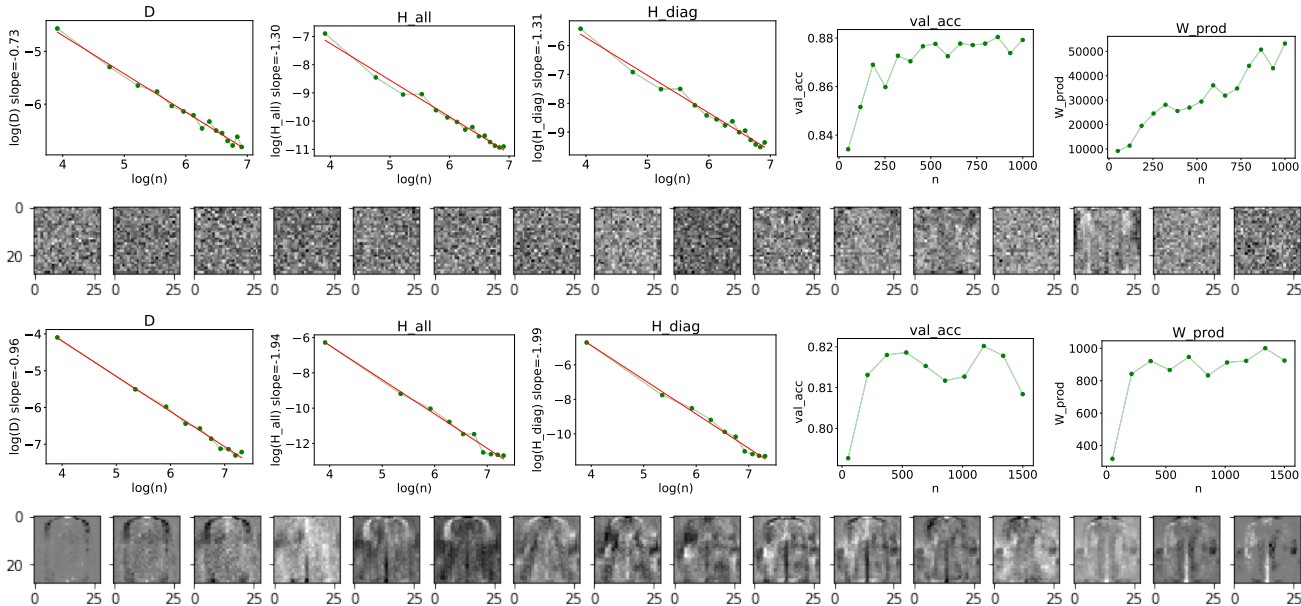

Figure 3: Non-regularized Fashion MNIST (first two rows), regularized Fashion MNIST (second two rows). Description is the same as in Fig. 2

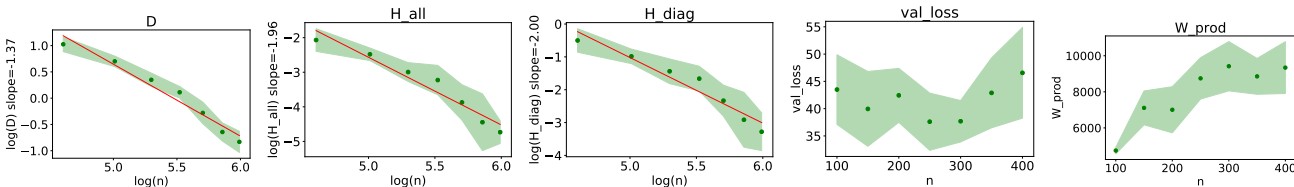

Figure 4: Non-regularized Boston Housing dataset

consider the expression for the variance and explicitly compute a correlation between $\xi^i \Delta_i'(t(\xi))$ and $\xi^j \Delta_j'(t(\xi))$. However, we were unsuccessful in doing so. Another approach is to take one more term in the expansion: $\mathcal{O}(r^5)$ – it will make the previous term with $p^3$ go away, leaving only terms $p^2/n$ and $p/n^2$, as the difference expressions in the variance will cancel out. Another idea is to use a Taylor series $\mathrm{Var}\Delta = \sum_{k=0}^{\infty} \frac{T_k}{k!} r^k$ (existing in principle). Next, we explicitly bound the remainder since we have a global bound on $D_k \leq D$. Finally, if we expand around the mean $\mu = \mathbb{E}x$: $\mathbb{E}f(x) = f(\mu) + \frac{1}{2}\Theta_{\pm}(1)(f_{ii}'', \mathrm{Var}x)[2]$, we get a remainder $p/n$ right away, but need to compute $f(\mu)$ which is the network at a modified input $(x - px)$.

# 5 Probabilistic Guarantees on The Fault Tolerance Using Tail Bounds

Under our assumptions, $\mathrm{Var}\Delta \sim \sum_l \frac{C_l p_l}{n_l}$. The constant $C_i$ comes from Theorem 1 (or its more computationally tractable form in Add. Coroll. 2). Since we know that the error superposition is linear (Additional Proposition 7), we sum the individual layer terms, hence $\sum_l$. Proposition 4 motivates the inverse dependency on $n_l$.

**Median trick.** Suppose that we have a random variable $X$ satisfying $\mathbb{P}\{X \geq \varepsilon\} < 1/3$. Then we create $R$ independent copies of $X$ and calculate $X = median\{X_i\}_{i=1}^{R}$. Then $\mathbb{P}\{X \geq \varepsilon\} < (1/3)^{R/2}$ because in order for the median to be larger than value $\varepsilon$, at least half of its arguments must be larger than $\varepsilon$, and all $X_i$ are independent. Thus $R = \mathcal{O}(\log 1/\delta)$ in order to guarantee $(1/3)^{R/2} < \delta$. This is a standard technique.

**Proposition 5.** *A neural network under assumptions 1-3 is $(\varepsilon, \delta)$-fault tolerant for $t = \varepsilon - \mathbb{E}\Delta_L > 0$ with $\delta = \delta_0 + t^{-2}\mathrm{Var}\Delta_L$ for $\mathbb{E}\Delta$ and $\mathrm{Var}\Delta$ calculated by Theorem 1 and $\delta_0$ from Proposition 3.*

*Proof.* We apply the Taylor expansion from Theorem 1. We directly apply the Chebyshev's inequality for $X = \Delta$:

$$\delta = \mathbb{P}\{X \geq t + \mathbb{E}X\} \leq \frac{\text{Var}X}{t^2}$$

□

## 5.1 Properties of The Fault Tolerance

**Proposition 6.** *Suppose that a $C^2$ network is at a stationary point after training: $\mathbb{E}_x \nabla_W L(x) = 0$. Then in the first order on $p, q$, $\mathbb{E}_x \mathbb{E}_\xi \Delta_{L+1} = 0$*

*Proof.* Consider the quantity in question, use Additional Proposition 9 for it and apply $\mathbb{E}_x$ on both sides:

$$\mathbb{E}_\xi \Delta_{L+1} = \mathbb{E}_x \left( \frac{\partial L}{\partial y} \mathbb{E}_\xi \Delta \right) = -p \left( \mathbb{E}_x \frac{\partial L}{\partial W}, W \right)$$

Now since we know that $\mathbb{E}_x \frac{\partial L}{\partial W} = 0$, the linear term is 0

□

**Additional Proposition 7.** *(Linearity of error superposition for small p limit) Consider a network $(L, W, B, \varphi)$ with crashes at each layer with probability $p_l$, $l \in \overline{0, L}$. Then in the first order on $p$, the total mean or variance of the error at the last layer is equal to a vector sum of errors in case crashes were at a single layer*

$$\mathbb{E}\Delta_L^{p_0,\ldots,p_L} = \mathbb{E}\Delta_L^{p_0} + \ldots + \mathbb{E}\Delta_L^{p_L}$$
$$\text{Var}\Delta_L^{p_0,\ldots,p_L} = \text{Var}\Delta_L^{p_0} + \ldots + \text{Var}\Delta_L^{p_L}$$

*Proof.* Consider each layer $i$ having a probability of failure $p_i \in [0, 1]$ for $i \in \overline{0, L}$.

In this proof we utilize Assumption 1. We write the definition of the expectation with $f(p, n, k) = p^k (1-p)^{n-k}$ being the probability that a binary string of length $n$ has a particular configuration with $k$ ones, if its entries are i.i.d. Bernoulli $Be(p)$. Here $S_l$ is the set of all possible network crash configurations at layer $l$. Each configuration $s_l \in S_l$ describes which neurons are crashed and which are working. We have $|S_l| = 2^{n_l}$.

$$\mathbb{E}\Delta_L^{p_0,\ldots,p_L} = \sum_{s_0 \in S_0} \cdots \sum_{s_L \in S_L} f(p_0, |s_0|, N_0) \ldots f(p_L, |s_L|, N_L)(\hat{y}_L - y_L)$$

where

$$\hat{y}_L = (\quad W_L \varphi(\ldots(\varphi(W_1(\quad x \odot \xi_0) \quad + b_1) \odot \xi_1) \quad \ldots) \odot \xi_{L-1} \quad + b_L) \odot \xi_L$$
$$y_L = \quad W_L \varphi(\ldots(\varphi(W_1 \quad x \quad + b_1) \quad \ldots) \quad + b_L$$

We utilize the fact that the quantity $f(p, n, k) = p^k (1-p)^{n-k} = p^k + \mathcal{O}(p^{k+1})$. Only those sets of crashing neurons $(s_0, \ldots, s_L)$ matter in the first order, which have one crash in total (in all layers). We denote it as $s_k^1 \in S_k$. Therefore we write

$$\mathbb{E}\Delta_L^{p_0,\ldots,p_L} = \sum_{s_0^1 \in S_0} f(p_0, 1, N_0)(\hat{y}_L - y_L) + \ldots + \sum_{s_L^1 \in S_0} f(p_L, 1, N_L)(\hat{y}_L - y_L)$$

This is equivalent to a sum of individual layer crashes up to the first order on $p$:

$$\mathbb{E}\Delta_L^{p_0,\ldots,p_L} = \mathbb{E}\Delta_L^{p_0} + \ldots + \mathbb{E}\Delta_L^{p_L}$$

The proof for the variance is analogous.

□

# 6 Algorithm for Certifying Fault Tolerance

In case if we consider the RHS quantities in Theorem 1 averaged over all data examples $(x, y^*)$, then the Algorithm 1 from the main paper would give a guarantee *for every example*: it will guarantee that $\mathbb{P}_{x,\xi}[\Delta \geq \mathbb{E}\Delta + t]$ is small. Indeed, if we know that $\mathbb{E}_x \text{Var}\Delta$ is small, we know that the total variance $\text{Var}_{x,\xi}\Delta = \mathbb{E}_x \text{Var}_\xi \Delta + \text{Var}_x \mathbb{E}_\xi \Delta$ (by the law of total variance) is small as well. Indeed, the second term $\text{Var}_x \mathbb{E}_\xi \Delta$ bounded by $\sup_x \mathbb{E}_\xi \Delta$ which we assume to be small. In case if it is not small, it is unsafe to use the network, as even the expectation of the error is too high.

Given a small total variance $\text{Var}_{x,\xi}\Delta$, we apply, as in Proposition 5 a Chebyshev's inequality to the random variable $\Delta$ over the joint probability distribution $x, D|x$. This will give $\mathbb{P}_{x,\xi}[\Delta \geq \mathbb{E}\Delta + t] \leq t^{-2}\text{Var}_{x,\xi}\Delta$. This probability indicates how likely it is that a network with crashes and random inputs will encounter a too high error.

Median aggregation of $R$ copies works for the input distribution as well. Denote "$y_i(x)$ is bad" as the event that the loss exceeds $\varepsilon$ for the i'th copy. We denote $[True] = 1$ and $[False] = 0$ (Iverson brackets). Now, $\delta = P_{x,\xi}(Med(y_i(x))$ bad$) = \mathbb{E}_x\mathbb{E}_\xi[$at least half of $y_i$ are bad$]$. Since the inner probability can be bounded as $\leq t^{-2}\text{Var}\Delta(x)\exp(-R)$, taking an expectation over $\mathbb{E}_x$ results in the quantity discussed before, $\mathbb{E}_x\text{Var}\Delta$.

We note that it would not be possible to consider $y_{L+1}$ to be the total loss, as it makes quantities such as $\partial L/\partial W^{ij}(x, W)$ ill-defined, as they depend on $x$ as well.

## 6.1    Analysis of the Algorithm 1

The algorithm consists of the main loop which is executed until we can guarantee the desired $(\varepsilon, \delta)$-fault tolerance. It trains networks, and upon obtaining a good enough network with $\delta < 1/3$, it repeats the network a logarithmic number of times. We note that the part on $q$ and $\delta_0$ is not strictly required to guarantee fault tolerance. Rather, satisfying these conditions is a natural necessary conditions to satisfy the more strict ones (on $R_3$, $\mathbb{E}\Delta$ and $\delta$). These conditions are necessary because, by AP5, continuous limit implies that the $q$ is reasonable.

**Space requirements.**    After each iteration of the main loop of the algorithm, the previous network can be deleted as it is no longer used. Therefore, each new iteration does not require any additional space. We only need to store the network itself and the computational graph for $\text{Var}\Delta$, which depends on network's gradients. The total space complexity is then $\mathcal{O}\left(\sum_{l=1}^{L} n_l \times n_{l-1}\right)$ which is just the space to store the network's weights.

**Time and neurons requirements.**    Each iteration trains the network and then performs computations which involve only 1 forward pass, so the bottleneck is still the training stage which we assume can be done in time $\mathcal{O}(T)$. Now let us calculate the number of iterations. New iteration is requested in case if one of the conditions is not satisfied. We assume the loss to be bounded in $[-1, 1]$ by D1. We analyze each of the possible non-satisfied conditions:

1. $q < 10^{-2}$. In that case the target value of $\max|W_i|/\min|W_i| > 100$ which is many times greater than the loss itself (by assumption $\omega \in [-1, 1]$). In this case the algorithm increases the regularization parameter $\mu$ to make $q$ smaller. We assume that we double $\mu$ each time. So by setting $\mu^* = 1$ the network will definitely achieve $q > 10^{-2}$. Therefore the number of doublings of $\mu$ is at most $\log 1/\mu^0$. Since $\mu^0$ is represented as a float, $\log 1/\mu^0 = \mathcal{O}(1)$

2. $\delta_0 > 1/3$. Since $\delta_0 = \exp(-n_l q d_{KL}(\alpha|p_l))$, it will decay exponentially with the decay of $n_l$. In order to make $\delta_0 < 1/3$, we need to set $n_l q d_{KL}(\alpha|p_l) > 2$ which leads to $n \geq \frac{200}{d_{KL}(\alpha|p_l)}$. Since both $\alpha$ and $p$ are small, we take only the first-order term in $d_{KL}(\alpha|p) = \alpha \log \alpha/p + (1-\alpha)\log(1-\alpha)/(1-p)$. The last term here is $\sim p - \alpha \geq -\alpha$. In case if $\alpha \geq e^2 p^6$, $\log \alpha/p \geq 2$ and $d_{KL}(\alpha|p) \gtrsim \alpha$. Now, $\alpha$ needs to be sufficiently small in order to guarantee the second-order term in the Taylor expansion $\alpha^2 D_{12} \ll 1$[7]. Therefore, $\alpha^2 \sim 1/D_{12}$ and $n \sim 200 D_{12}$. If the algorithm increases $n$ by a constant amount, it will need to take $\mathcal{O}(D_{12})$ operations

3. $R_3 > C$. In this case, we increase $\psi$. Since this directly influences $R_3$ via regularization, doubling $\psi$ every time results in a similar $\mathcal{O}(1)$ performance as in the analysis for $q$ and $\mu$. Since $R_3 \approx C$ is the number of changes that the weights make, which is $> 1$, and the loss is bounded by 1.

4. $\delta = \varepsilon^{-2}\text{Var}\Delta \approx \varepsilon^{-2}\frac{C_l p_l}{n_l} > 1/3$ (see Section 5) for $C_l$ dependent on the function approximated by the NN and the continuous limit. Therefore, $n \sim \mathcal{O}(C_l p_l/\varepsilon^2)$.

The total number of iterations is therefore $\mathcal{O}(D_{12} + C_l p_l/\varepsilon^2)$ for $C_l = n_l\text{Var}\Delta/p_l$ for some $n_l$ (this is now a property of the function being approximated and the continuous limit). We note that the constants $1/3$ and $10^{-2}$ are chosen for the simplicity of the proof. The asymptotic behavior of the algorithm does not depend on them, as long as they are constant.

---

[6]For $e$: $\log e = 1$

[7]Technically, here we silently implied that $e^2 p \ll 1/\sqrt{D_{12}}$ in order to make $\alpha \geq e^2 p$, which means that we *cannot* implement a function with too high second derivatives in neuromorphic hardware with a constant $p$, not matter how many neurons we take. Intuitively, this happens because for such a function, even a failure of $e^2 p$ fraction of neurons, which is a reasonable expectation, is too large to begin with. We assume that we are fitting a function which is not like that. If we encounter this, we will see that by having a too high $\mathbb{E}\Delta$ and the algorithm will output `infeasible`

**Correctness: guarantee of robustness.** Now we analyze correctness. We note that the algorithm is not guaranteed to find a good trade-off between accuracy and fault-tolerance. Up to this point, there is no complete theory explaining the generalization behavior of neural networks or their capacity. Therefore, we cannot give a proof for a sufficient trade-off without discovering first the complete properties of NNs capacity. We only show that the algorithm can achieve fault tolerance.

First, the condition on $\mathbb{E}\Delta$ and $\mathrm{Var}\Delta$ implies that the first-order terms in the expansion from T1 are small enough. Now we argue that the remainder is small as well.

The condition $R_3 < C$ implies that discrete function is smooth enough to apply $\int |W_t'(t,t')dtdt'| \approx C_1 < R_3 < C$ as well as $\int |W_{t'}'(t,t')|dtdt' \approx C_2 < R_3 < C$. This means that the integral is small, which allows to bound the Riemann remainder $R_3/n_l$ from the proof of AP4. This implies that there exist a continuous network $NN_c$ such that the approximation error $A$ from AD2 is small. Here, the right metric is $R_3/n_l \approx C/n_l \ll 1$. The number of changes $C$ must be less than a number of neurons at a layer $n_l$.

Now, the remainder depends on the operator derivative bound $D_{12}$. By the Assumption 3 (part on $D_k$), they are small. Below we describe why this part holds.

**Experimental evidence for $D_k$ being small.** First, since we explicitly have a term $R_1$ for $\mathrm{Var}\Delta \geq \left(\partial y/\partial y_l^i\right)^2 y_l^i$ (AC2) in the regularizer, the first-order derivatives $D_1$ would be small for each layer by design of our algorithm. Next, second-order derivatives $\partial^2 y_L/\partial W_l^{ij}\partial W_l^{i'j'}$ are found to be small in experimental studies of the Hessian [11]. Since derivatives w.r.t. $y_l$ can be expressed via the derivatives w.r.t. weights by AP8, and the continuous limit holds, $D_2$ is small as well.

> **General considerations for $D_k$ being small.** Note, we can always renormalize $|x(t)| \leq 1$ and $|y(t)| \leq 1$ by rescaling the input and output weights. For the input layer, $D_{12}$ can be bounded via the properties of the ground truth function $y^*$, if $y$ approximates $y^*$ well. However, it is known that it can be false: the network can have a sufficient accuracy on the training dataset, but have much larger output-input derivatives ($d[out]/d[in]$) than the ground truth function. Specifically, in the task of image recognition, we (humans) assume the problem to be quite smooth: a picture of a cat with a missing ear or whiskers is still a picture of a cat. However, it was shown [13] that modern CNNs use *non-robust* features. This implies that CNNs are much more sensitive to small changes in input $x$, in contrast to the smooth ground truth function $y^*$ that we want it to learn, making the bound for the derivative $D_1$ large.
>
> For the hidden layers, it is possible that the continuous limit has large derivatives. For example, we can first apply an injective transformation $\mathcal{F}$ with high derivatives, and then apply the inverse transform implemented as another neural network $\mathcal{F}^{-1}$ existing by AP3[a]. Then, we have an overall smooth (identity) operator $y_L$ which, however, consists of two very non-smooth parts.
>
> We list the following approaches that potentially could resolve this issue theoretically rather than experimentally. First, we can consider the infinite depth limit [23]. This would allow to have regularities throughout the network' layers. Another approach is to study mathematically the ways that an operator can be decomposed into a hierarchical composition of operators. For example, for images, a natural decomposition of the image classification operator could first detect edges, then simple shapes, then groups them into elements pertinent to specific classes and then detects the most probable class [26]. At each stage, only robust features are used. Thus, for such a decomposition, output-hidden derivatives would be reasonable as well as the output-input ones: indeed, the decision to recognize a cat would not change significantly if some internal hidden layer features (ears or whiskers) are not present. Interestingly, just enforcing the continuous limit seems to make features more robust in our experiments, see hidden layer weights on Figures 2 and 3. Without regularization, the weights seem noisy, there are a lot of unused neurons, and even the used ones contain noisy patterns. In contrast, continuity-regularized networks seem to have first-layer weights similar to the input images. This could be an interesting research direction to continue.
>
> We note, however, that such a study is not connected to the fault tolerance anymore, as it is a fundamental investigation into the properties of the hierarchical functions that we want to approximate, and into the properties of the neural networks which we can find by gradient descent.

---

[a]The idea to construct $y = \mathcal{F} \circ \mathcal{F}^{-1}$ with $\mathcal{F}^{-1}$ implementable by a neural network is taken from [8]

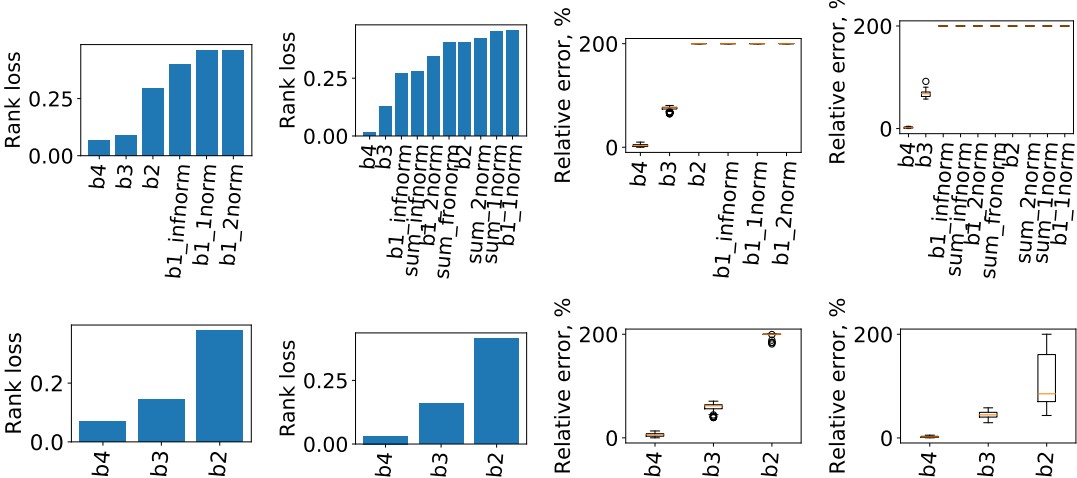

Figure 5: Boston-trained networks. Rank loss (worst value 0.5) of ordering networks using bounds (left 4) and relative error in predicting the value of $\Delta$ (right 4). Mean of the error $\mathbb{E}\Delta$ (top) and standard deviation (bottom). Variable input on a fixed network (1st and 3rd columns) and variable network with fixed input (2nd and 4th columns).

# 7 Experimental Evaluation

Experiments were performed on a single machine with 12 cores, 60GB of RAM and 2 NVIDIA GTX 1080 cards running Ubuntu 16.04 LTS and Python/Tensorflow/Keras. We test the proposed bounds on two datasets as a proof of concept: the Boston Housing dataset (`https://www.cs.toronto.edu/~delve/data/boston/bostonDetail.html`) (regression) and the MNIST dataset (classification) (`http://yann.lecun.com/exdb/mnist/`). In addition, we use the Fashion MNIST dataset (classification, `https://www.kaggle.com/zalando-research/fashionmnist`). We use standard pre-trained networks from Keras (VGG, MobileNet, ..., `https://keras.io/applications/`).

For the Dropout experiment $p < 0.03$ is used as a threshold after which no visible change is happening. We use the unscaled version of dropout[8].

In the experiments, we use computationally tractable evaluations of the result given by T1. The "b1 bound" is the Spectral bound from P2. The "b2 bound" corresponds to AP2. The "b3 bound" corresponds to first-order terms from T1, or the Additional Corollary 2, and the "b4 bound" corresponds to an exact evaluation of single-neuron crashes (taking $\mathcal{O}(n_l)$ forward passes).

## 7.1 Boston-trained networks with different initialization: rank loss with experimental error (additional)

We compare sigmoid networks with $N \sim 50$ trained on Boston Housing dataset (see `ErrorComparisonBoston.ipynb`). We use different inputs on a single network and single input on different networks. We compare the bounds using rank loss which is the average number of incorrectly ordered pairs. The motivation is that even if the bound does not predict the error exactly, it could be still useful if it is able to tell which network is more resilient to crashes. The second quantity is the relative error of $\Delta$ prediction, which is harder to obtain. Experimental error computed on a random subset $S'$ of all possible crashed configurations $S$ with $|S| = 2^{20} \sim 10^6$ is used as ground truth, with $|S'|$ big enough to make the expectation and variance results not change from one launch to another. The results are shown in Figure 5. Even on this simple dataset we have only b3 and b4 giving meaningful results both for rank loss and relative error of error. The results show that b4 is always better than b3, as expected. The same is done for random networks.

---

[8]We note that it is the same as the scaled version up to the first order

## 7.2 Comparison on random networks (additional)

The same setup as for the Different Dropout Comparison experiment in the main paper is done for random networks with $N \sim 20$, ReLU activation function, see `ErrorComparisonRandom.ipynb`. The results are shown in Figure 6 and they are qualitatively the same as for Boston dataset.

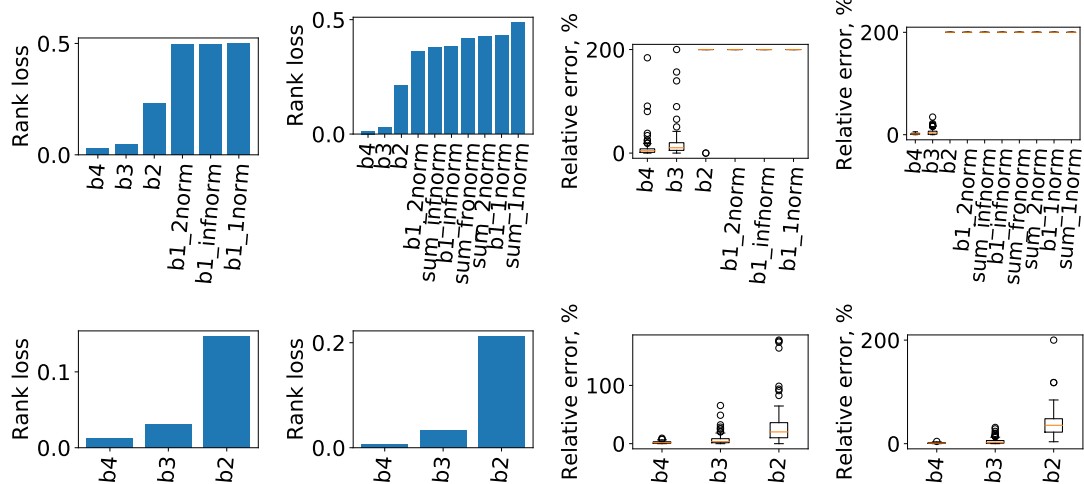

Figure 6: Random networks. Rank loss (worst value 0.5) of ordering networks using bounds (left 4) and relative error in predicting the value of $\Delta$ (right 4). Mean of the error $\mathbb{E}\Delta$ (top) and standard deviation (bottom). Variable input on a fixed network (1st and 3rd columns) and variable network with fixed input (2nd and 4th columns).

## 7.3 Additional charts

Table 2 presents the results of `ConvNetTest-ft.ipynb`.

Figure 9 presents the results from `ComparisonIncreasingDropoutMNIST.ipynb`.

Figure 10 shows results from `Regularization.ipynb`

Figure 8 shows how the algorithm's (Main paper, Algorithm 1, `TheAlgorithm.ipynb`) state evolves over iterations. Figure 7 shows the resulting distribution of the error $\Delta$ at the output layer for the final network given by the algorithm.

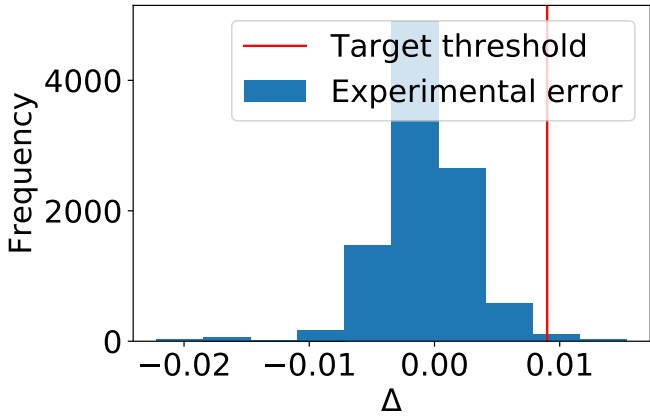

Figure 7: The distribution $\Delta$ of the error in the output for the network obtained using Algorithm 1 from the main paper

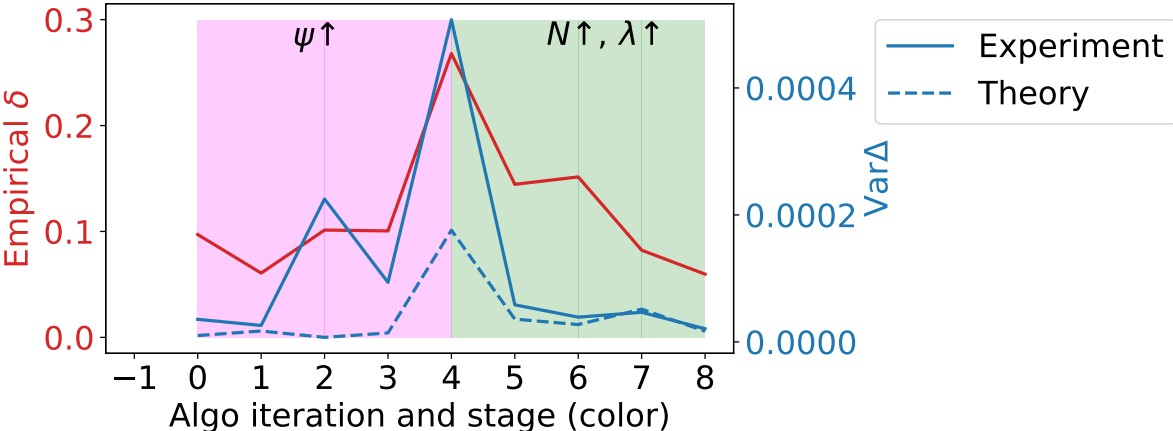

Figure 8: Evolution of the experimental error probability and the variance of the error over algorithm's iterations. There are two stages. In the first stage (red, iterations $\leq 4$) the algorithm increases continuity of the network via increasing $\psi$. In the second stage (green, iterations $\geq 5$) the algorithm increases the number of neurons $n_1$ and the regularization parameter $\lambda$. First, it can be seen that at first stages the network is not continuous enough, as the algorithm makes it more continuous. This leads to an increase in the empirical probability $\delta$ of the network outputting a loss $> \varepsilon$ and in the increase in the gap between Theoretical Var$\Delta$ and Experimental Var$\Delta$. This happens because there are not enough neurons in the network. Later, as the number of neurons increases, the gap becomes smaller and the empirical probability decreases. Note that the first network (at iteration 0) empirically satisfies our fault tolerance guarantee. Nevertheless, we do not have a proof for such a network because it is not continuous enough. Therefore, in order to guarantee robustness, we need to proceed with the iterations.

| Model | log Var$\Delta$ | log[Parameters/Layers] | log[Parameters] | Layers |
|---|---|---|---|---|
| VGG16 | -18.7 | 15.6 | 18.7 | 23 |
| VGG19 | -18.4 | 15.5 | 18.8 | 26 |
| MobileNet | -9.1 | 10.7 | 15.3 | 93 |

Table 2: Comparison of bigger convolutional networks when there are faults at every layer with $p = 10^{-2}$

## 7.4 Error superposition (testing AP7, additional)

We test a random network with $L = 4$ on random input (see `ErrorAdditivityRandom.ipynb`). The error is computed on subsets of failing layers $a, b$. Then all pairs of disjoint subsets are considered, and the relative error of $\Delta$ estimation using linearity is computed as $\|\Delta_{a \cup b} - \Delta_a - \Delta_b\| / \|\Delta_{a \cup b}\|$. The results (see Figure 11) show that this relative error is only few percent both for mean and variance, with better results for the mean.

## 7.5 Testing AP9 (additional)

The result predicts the expected error $\mathbb{E}_x \mathbb{E}_\xi \frac{\partial L}{\partial y} \Delta$ to decay with the decay of the gradient of the loss. We tested that experimentally on the Boston dataset using sigmoid networks (see `ErrorOnTraining.ipynb`) and note that a similar result holds for ReLU[9]. The results are shown in Figure 12. The chart shows first that the experiment corresponds to b4, and b3 is close to b4. b3 is also equal to the result of AP9, both of which decay, as predicted.

## 7.6 Practical bounds

---

[9]ReLU$\notin C^1$. However only a small percentage of neurons are near 0 thus approximation works for most of them.

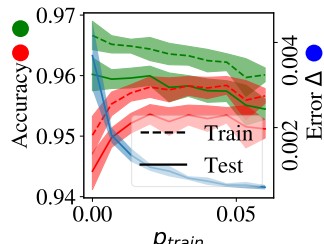

Figure 9: Comparison of networks trained with different dropout. Shown: accuracy and Var$\Delta$ plots for different train dropout probability. Green curve shows accuracy of correct network, red shows accuracy for the crashing network. Dashed line show train dataset and solid represent test dataset. Variance of $\Delta$ estimated by b4 shown in orange, by b3 in blue. Error bars are standard deviations in 10 repetitions of the experiment.

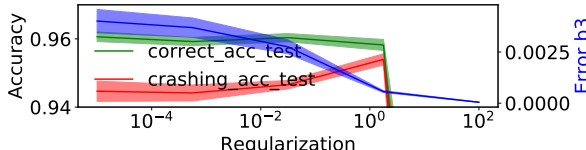

Figure 10: Horizontal axis represents increasing regularization parameter. Left vertical axis represents accuracy and has test crashing network accuracy (red) and correct network accuracy (green). Right axis shows variance of the error Var$\Delta$ estimated by bound b3 used as a regularizer. Error bars are standard deviations in 5 repetitions of the experiment.

**Additional Proposition 8.** *For neural network we have for any weight matrix $W_l$ and input to the $l$'th layer $y_{l-1}$:*

$$\sum_{i,j} \frac{\partial y}{\partial W_l^{ij}} W_l^{ij} = \sum_j \frac{\partial y}{\partial y_{l-1}^j} y_{l-1}^j$$

*The equality also holds for one particular $y_{l-1}^j$:*

$$\sum_i \frac{\partial y}{\partial W_l^{ij}} W_l^{ij} = \frac{\partial y}{\partial y_{l-1}^j} y_{l-1}^j$$

542

543  *Proof.* Fix some layer $l$. The output of the network $y$ depends on the weight matrix $W_l$ and on the input to the
544  $l$'th layer $y_{l-1}$. However we note that it only depends on their product and not on these quantities separately.
  Therefore we write $y = y_L(W_l, y_{l-1}, ...) = \eta(W_l y_{l-1})$ and denote $x = y_{l-1}$, $W = W_l$ and $z = Wx$:

$$y = \eta(Wx)$$

Now we take one

$$\frac{\partial y}{\partial x_j} = \sum_i \frac{\partial \eta}{\partial z_i} \frac{\partial z_i}{\partial x_j}$$

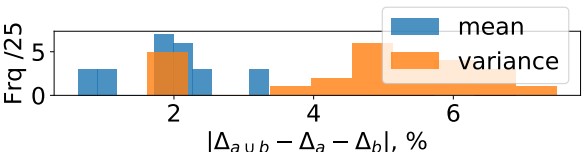

Figure 11: Distribution of relative error in the predicted $\Delta$ in percent for 25 subset pairs. Distribution for the mean is shown in blue, for the variance in orange.

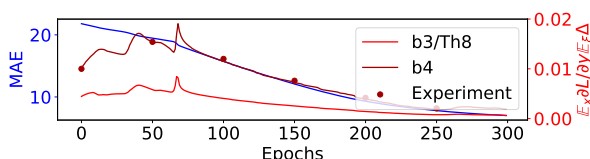

Figure 12: Decay with training of the loss (blue line, left vertical axis) and the error $\mathbb{E}_x \frac{\partial L}{\partial y} \mathbb{E}_\xi \Delta$ (red, right axis) together with bounds b3 and b4 predictions (red and brown curves) and experimental values (dots). "Th8" means AC2

Since $z_i = \sum_k W_{ik} x_k$, $\frac{\partial z_i}{\partial x_j} = W_{ij}$. Therefore we plug that in:

$$\frac{\partial y}{\partial x_j} = \sum_i \frac{\partial \eta}{\partial z_i} W_{ij}$$

And multiply with $x_j$:

$$\frac{\partial y}{\partial x_j} x_j = \sum_i \frac{\partial \eta}{\partial z_i} W_{ij} x_j$$

Now we compute $\frac{\partial y}{\partial W_{ij}}$ using that in vector $z$ only $z_i$ depends on $W_{ij}$ and also that $z = Wx$:

$$\frac{\partial y}{\partial W_{ij}} = \frac{\partial \eta}{\partial z_i} \frac{\partial z_i}{\partial W_{ij}} = \frac{\partial \eta}{\partial z_i} x_j$$

Then we multiply it by $W_{ij}$ and sum over $i$:

$$\sum_i \frac{\partial y}{\partial W_{ij}} W_{ij} = \sum_i \frac{\partial \eta}{\partial z_i} x_j W_{ij}$$

And now we note that the expressions for $\frac{\partial y}{\partial x_j} x_j$ and $\sum_i \frac{\partial y}{\partial W_{ij}} W_{ij}$ are exactly the same.

Differentiating the above expression gives a connection between second derivatives w.r.t weights and activations. $\qquad\square$

**Additional Corollary 2.** *The error expressed in weights in the first order is as in T1:*

$$\mathbb{E}_\xi \Delta_L^l = -p \sum_{i,j} \frac{\partial y}{\partial W_l^{ij}} W_l^{ij}$$

$$\mathrm{Var}\Delta_L^l = p \sum_i \sum_{j,k} \frac{\partial y}{\partial W_l^{ji}} \frac{\partial y}{\partial W_l^{ki}} W_l^{ji} W_l^{ki}$$

*Where $\Delta_L^l$ is the error in case of neurons failing at layer $l$. Layer $0$ means failing input.*
*In other words, both mean and variance are defined by a tensor $X_l = W_l \odot \frac{\partial y}{\partial W_l}$, $z_l^j = -\sum_i X_l^{ij}$:*

$$E\Delta_L^l = p \sum_i z_l^i, \ \mathrm{Var}\Delta_L^l = p \sum_i (z_l^i)^2$$

*Proof.* According to Theorem 1 the expression for mean and variance are:

$$\mathbb{E}\Delta \approx -p(\nabla y(x), x) = -p \sum_i \frac{\partial y}{\partial x_i} x_i$$

$$\mathrm{Var}\Delta \approx ((\nabla y(x))^2, x^2) = p \sum_i \left( \frac{\partial y}{\partial x_i} x_i \right)^2$$

Then by Additional Proposition 8 the expression used can be rewritten as (note the index swap $i \leftrightarrow j$):

$$\frac{\partial y}{\partial x_i} x_i = \sum_j \frac{\partial y}{\partial W_{ji}} W_{ji}$$

Then for the mean it is:

$$\mathbb{E}\Delta = -p \sum_i \sum_j \frac{\partial y}{\partial W_{ji}} W_{ji}$$

and the indices can be swapped again for the sake of notation. For the variance we write the square as the inner sum repeated twice with different indices $k$ and $j$:

$$\mathrm{Var}\Delta = p \sum_i \sum_j \sum_k \frac{\partial y}{\partial W_{ji}} W_{ji} \frac{\partial y}{\partial W_{ki}} W_{ki}$$

This gives the first statements. Then we notice that both expressions depend on the values of tensor $\frac{\partial y}{\partial W} \odot W$ but not on $W$ or $\frac{\partial y}{\partial W}$ alone. We therefore define $X = W \odot \frac{\partial y}{\partial W}$ and $z_l^j = -\sum_i X_{ij}$ (sum over the first index) and rewrite

$$\mathbb{E}\Delta = -p \sum_i \sum_j X_{ij} = p \sum_j z^j$$

$$\mathrm{Var}\Delta = p \sum_{ijk} X_{ji} X_{ki} = p \sum_i \left( \sum_j X_{ji} \right)^2 = p \sum_i (-z_i)^2$$

□

**Additional Proposition 9.** *For a neural network with $C^1$ activation function we have for a particular input $x$ in the first order as in T1:*

$$\mathbb{E}_\xi \Delta_{L+1} = -p(\nabla_W L, W)$$

*Proof.* Take the expression $A = \mathbb{E}_\xi \Delta_{L+1} = \left( \frac{\partial L}{\partial y}, \mathbb{E}_\xi \Delta \right)$ and consider $\mathbb{E}_\xi \Delta$. By Additional Corollary 2 we have $\mathbb{E}_\xi \Delta = -p \sum_{ij} \frac{\partial y}{\partial W_{ij}} W_{ij}$

Therefore,

$$A = -p \left( \frac{\partial L}{\partial y}, \sum_{ij} \frac{\partial y}{\partial W_{ij}} W_{ij} \right) = -p \sum_{ij} W_{ij} \left( \frac{\partial L}{\partial y}, \frac{\partial y}{\partial W_{ij}} \right)$$

By the chain rule for a single input $x$ we have

$$\frac{\partial L}{\partial W_{ij}} = \left( \frac{\partial L}{\partial y}, \frac{\partial y}{\partial W_{ij}} \right)$$

We plug that back in:

$$A = -p \sum_{ij} \frac{\partial L}{\partial W_{ij}} W_{ij}$$

This is the exact statement from the proposition. □

# 8 Unused extra results

559 **Skip-connections.** If we consider a model with skip-connections (which do not fit Definition 1) with faults at
560 every node, we expect that an assumption similar to A3 would lead to a result similar to T1. However, we did not
561 test our theory on models with skip-connections.

562 **Another idea for fault tolerance.** In case if we know $p$ exactly, we can compensate for $\mathbb{E}\Delta$ by multiplying
563 each neuron's output by $(1-p)^{-1}$. In this way, the mean input would be preserved, and the network output will
564 be unbiased. However, this only works in case if we know exactly $p$ of the hardware.

**Additional Proposition 10** (Variance for bound b1, not useful since the bound is not tight. Done in a similar
manner as the mean bound in [9])**.** *The variance of the error* $\mathrm{Var}\Delta$ *is upper-bounded as*

$$\mathrm{Var}\Delta \leqslant \sum_{i=1}^{n} a_i \prod_{j=i+1}^{n} b_j$$

*for*

$$
\begin{aligned}
a_L &= C_L^2 p_L(\alpha_L + p_L\beta_L) + 2C_L p_L(1-p_L)\beta_L\mathbb{E}\Delta_{L-1}, \\
b_L &= (1-p_L)(\alpha_L + (1-p_L)\beta_L) \\
C_l &= \max\{|y_{li}|\} \\
\alpha_L &= \max\{(w^{(L+1)}, w^{'(L+1)})\} \\
\beta_L &= \max\{\|w^{(L+1)}\|_1\|w^{'(L+1)}\|_1 - (w^{(L+1)}, w^{'(L+1)})\}
\end{aligned}
$$

565 *Proof.* In the notation of [9], consider $|\Delta_L| = |F_{\mathrm{neu}} - F_{\mathrm{fail}}| = |\sum\limits_{i=1}^{N_L} w_i^{(L+1)}(y_j^{(L)} - \hat{y}_j^{(L)}\xi_j^{(L)})|$, $\mathbb{E}\Delta_L \leqslant p_l\|w^{(L+1)}\|_1 C_L +$

566 $K\|w^{(L+1)}\|_1(1-p_L)\mathbb{E}\Delta_{L-1}$, where $\hat{y}_j^{(L)}$ is the corrupted output from layer $L$ and $K$ is the activation function

567 Lipschitz constant. Thus, $\mathbb{E}\Delta_L \leqslant \sum\limits_{l=1}^{L} p_l\|w^{(L+1)}\|_1 C_l K^{L-l} \prod\limits_{s=l+1}^{L}(1-p_s)\|w^{(s)}\|_1 = \sum\limits_{l=1}^{L} p_l\|w^{(l+1)}\|_1 C_l \prod\limits_{s=l+1}^{L} K(1-$

568 $p_s)\|w^{(s+1)}\|_1$

569 1. Variance: $\mathrm{Var}\Delta = \mathbb{E}(\Delta^2) - (\mathbb{E}\Delta)^2 \leqslant \mathbb{E}(\Delta^2)$, here we will calculate $\mathbb{E}\Delta^2$:

570 2. $\mathbb{E}\Delta_1^2 = \mathbb{E}\sum\limits_{j_1=1}^{N_1}\sum\limits_{j_2=1}^{N_1} w_{j_1}^{(2)}y_{j_1}^{(1)}(1-\xi_{j_1}^{(1)})w_{j_2}^{(2)}y_{j_2}^{(1)}(1-\xi_{j_2}^{(1)}) \leqslant f_1 C_1^2\overline{w_j^2} + C_1^2\frac{f_1^2-f_1}{N_1^2-N_1}\sum\limits_{j_1\neq j_2} w_{j_1}w_{j_2}$

571 $\frac{f_1^2-f_1}{N_1^2-N_1} \leqslant \frac{f_1^2}{N_1^2} = p^2.\ \sum\limits_{j_1\neq j_2} w_{j_1}w_{j_2} = \|w\|_1^2 - \|w\|_2^2.$

572 Therefore, $\leqslant \boxed{C_1^2 p\|w\|_2^2 + C_1^2 p^2(\|w\|_1^2 - \|w\|_2^2)}$

573 3. $\mathbb{E}\Delta_1\Delta_1' = \mathbb{E}\sum\limits_{j_1=1}^{N_1}\sum\limits_{j_2=1}^{N_1} w_{j_1}^{(2)}y_{j_1}^{(1)}(1-\xi_{j_1}^{(1)})w_{j_2}^{'(2)}y_{j_2}^{(1)}(1-\xi_{j_2}^{(1)}) \leqslant f_1 C_1^2\overline{w_i w_i'} + C_1^2\frac{f_1^2-f_1}{N_1^2-N_1}\sum\limits_{j_1\neq j_2} w_{j_1}w_{j_2}'$

574 $\leqslant \boxed{pC_1^2(w, w') + p^2 C_1^2(\|w\|_1\|w'\|_1 - (w, w'))}$

575 4. $\mathbb{E}\Delta_L^2 \leqslant \sum\limits_{i=1}^{N_L}(w_i^{(L+1)})^2\mathbb{E}(y_i^{(L)} - \hat{y}_i^{(L)}\xi_i^{(L)})^2 + \sum\limits_{i\neq j}^{N_L} w_i^{(L+1)}w_j^{(L+1)}\mathbb{E}\left[(y_i^{(L)} - \hat{y}_i^{(L)}\xi_i^{(L)})(y_j^{(L)} - \hat{y}_j^{(L)}\xi_j^{(L)})\right]$

576 $\leqslant \boxed{P_L\|w\|_2^2 + Q_L(\|w\|_1^2 - \|w\|_2^2)}$

577 5. $\mathbb{E}\Delta_L\Delta_L' \leqslant \sum\limits_{i=1}^{N_L} w_i^{(L+1)}w_i^{'(L+1)}\mathbb{E}(y_i^{(L)} - \hat{y}_i^{(L)}\xi_i^{(L)})^2 + \sum\limits_{i\neq j}^{N_L} w_i^{(L+1)}w_j^{'(L+1)}\mathbb{E}\left[(y_i^{(L)} - \hat{y}_i^{(L)}\xi_i^{(L)})(y_j^{(L)} - \hat{y}_j^{(L)}\xi_j^{(L)})\right]$

578 $\leqslant \boxed{P_L(w, w') + Q_L(\|w\|_1\|w'\|_1 - (w, w'))}$

579 6. $P_1 = pC_1^2$, $Q_1 = p^2 C_1^2$

580 7. $P_L = \mathbb{E}(y - \hat{y}\xi)^2 = \mathbb{E}y_i^2\mathbb{P}\{\xi = 0\} + K^2\mathbb{E}\Delta_{L-1}^2\mathbb{P}\{\xi = 1\} \leqslant C_L^2 p_L + K^2\mathbb{E}\Delta_{L-1}^2(1-p_L).$

581 8. $Q_L = \mathbb{E}(y - \hat{y}\xi)(y' - \hat{y}'\xi') \leqslant C_L^2\underbrace{\mathbb{E}\eta\eta'}_{\leqslant p_L^2} + (1-p_L)p_L KC_L\mathbb{E}(|\Delta_{L-1}| + |\Delta_{L-1}'|) + (1-p_L)^2 K^2\mathbb{E}\Delta_{L-1}\Delta_{L-1}'.$

9. Consider a recurrence $x_1 = a_1$, $x_n = a_n + b_n x_{n-1}$. Then $x_n = \sum\limits_{i=1}^{n} a_i \prod\limits_{j=i+1}^{n} b_j$

10. Define $\alpha_L = \max\{(w^{(L+1)}, w^{'(L+1)})\}$ and $\beta_L = \max\{\|w^{(L+1)}\|_1 \|w^{'(L+1)}\|_1 - (w^{(L+1)}, w^{'(L+1)})\}$. Then

$$\boxed{\mathbb{E}\Delta'_L \Delta_L, \ \mathbb{E}\Delta_L^2 \leqslant P_L \alpha_L + Q_L \beta_L}$$

11. Thus, $\mathbb{E}\Delta_L^2 \leqslant a_L + b_L \mathbb{E}\Delta_{L-1}^2$, where

$$a_L \ = \ C_L^2 p_L(\alpha_L + p_L \beta_L) + 2KC_L p_L(1 - p_L)\beta_L \mathbb{E}\Delta_{L-1}, \tag{1}$$
$$b_L \ = \ K^2(1 - p_L)(\alpha_L + (1 - p_L)\beta_L). \tag{2}$$

Therefore,

$$\boxed{\mathbb{E}\Delta_L^2 \leqslant \sum_{l=1}^{L} \left( C_l^2 p_l(\alpha_l + p_l \beta_l) + 2KC_l p_l(1 - p_l)\beta_l \mathbb{E}\Delta_{l-1} \right) \prod_{l'=l+1}^{L} K^2(1 - p_{l'})(\alpha_{l'} + (1 - p_{l'})\beta_{l'})}$$

$\square$

The goal of this proposition was to give an expression for the variance in a similar manner as it is done for the mean in [9]. However this proposition did not make it to the article because bound b1 was not showing any good experimental results.

# 9 Introduction into fault tolerance for neuromorphic hardware

**Neuromorphic hardware (NH).** Given the amount of processing power required for modern Machine Learning applications, emerging hardware technologies are nowadays reviving the *neuromorphic* project and its promise to cut off energy consumption of machine learning by several orders of magnitude [10]. Neuromorphic implementation of an NN is a physical device where each neuron corresponds to a piece of hardware, and neurons are physically linked forming weights. Thus, the computation is done (theoretically) at the speed of light, compared to many CPU/GPU cycles. The surge in performance could arguably even exceed the one that followed the switch from training on CPUs to training on GPUs and TPUs [3]. Recent results on neuromorphic computing report on concrete successes such as milliwatt image recognition [10] or basic vowel recognition using only four coupled nano-oscillators [22]. Since the components of a neuromorphic network are small [25] and unreliable [17], there are crashes in individual *neurons* or *weights* inside the network [25, 17]. They lead to a performance degradation. A failure within a neuromorphic architecture involved in a mission-critical application could be disastrous. Hence, fault tolerance in neural networks is an important concrete Artificial Intelligence (AI) safety problem [4]. In terms of fault tolerance, the unit of failure in these architectures is fine-grained, i.e., an individual *neuron* or a single *synapse*, with failure mode frequently being a complete crash. This is in contrast with the now classical case of a neural network as a software deployed on a single machine where the unit of failure is coarse-grained, i.e., the whole *machine* holding the entire neural network. For instance, in the popular distributed setting of ML, the so-called parameter-server scheme [16], the unit of failure is a *worker* or a *server*, but never a *neuron* or a *synapse*. Whilst very important, fine-grained fault tolerance in neural networks has been overlooked as a concrete AI safety problem.

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
