# OpenReview forum: "The Probabilistic Fault Tolerance of Neural Networks in the Continuous Limit"
_ICLR.cc/2020/Conference — Reject_

### Official Review · AnonReviewer1 · 2019-10-20
**Official Blind Review #1**

**Rating:** 1

**Review:**



Review: This paper considers the problem of dropping neurons from a neural network.  In the case where this is done randomly, this corresponds to the widely studied dropout algorithm.  If the goal is to become robust to randomly dropped neurons during evaluation, then it seems sufficient to just train with dropout (there is also a gaussian approximation to dropout using the central limit theorem called "fast dropout").

I think there are two directions I find interesting that are touched on by this paper.  One is the idea of dropping neurons as an adversarial attack, which I think has been studied empirically but not theoretically (to my knowledge).  However then it would be important to specify the budget of the attack - how many neurons can they remove and how precisely can they pick which neuron to remove?  Another would be studying the conditions for dropout to be useful as a regularizer (and not underfit), which are again somewhat understood experimentally but could deserve a more theoretical treatment.

However I don't think this paper solved a sufficiently clear problem and the motivation is somewhat confusing to me, especially when it seems like an analysis of dropout, and that isn't even mentioned until the 7th page.

Notes:
  -Paper considers loss of function from loss of a few neurons.

  -Idea is to study more rigorously the fault tolerance of neural networks to losing a subset of the neurons.

  -In terms of impact of the work, one thing to consider is that even if a normally or arbitrarily trained network doesn't have perfect fault tolerance to dropping neurons, a neural network *trained* with dropping networks could learn hidden states which become more fault tolerant.


Minor Comments:
  -I'm a bit unhappy with the argument about the brain losing neurons unless it has better referencing from neuroscience.  I imagine it's true in general but I wouldn't be surprised if some neurons were really essential.  For example squid have a few giant neurons that control propulsion.  It's just the first sentence so maybe I'm nitpicking.

  -  It also seems weird that the opening of the paper doesn't give more attention to dropout, since it's a well known regularizer and seems rather closely related conceptually.

  - In the intro it says neuromorphic hardware would pass information at the speed of light.  Is this really true?  My understanding is neuromorphic hardware would still generally use electrical or chemical signals but not pass things at the speed of light.

  - The citation format is not valid for ICLR.


**Experience Assessment:**

I have published in this field for several years.

**Review Assessment: Checking Correctness Of Derivations And Theory:**

I carefully checked the derivations and theory.

**Review Assessment: Checking Correctness Of Experiments:**

I carefully checked the experiments.

**Review Assessment: Thoroughness In Paper Reading:**

I read the paper thoroughly.

---

> ### Author Response · Authors · 2019-11-12
> **Our paper is about Fault Tolerance, not Dropout**
>
> Dear reviewer,
> Thank you for your time. It seems that several points on our end would require clarification with regards to your review.
>
> (Connection to Dropout and differences) Dropout is indeed connected to Fault Tolerance. It is in this context that this algorithm was invented [1]. Reducing overfitting came out as a bonus in Kerlirzin’s work and Dropout would later be rebranded as it was crucial in the revival of neural networks [2,3] from 2012 on.  However, our problem is different from Dropout since we are interested in calculating the error rather than in generalization properties.
>
> We note that Dropout indeed helps fault tolerance, but it cannot be seen as a solution to our problem: first, we do not know the true failure probability in hardware exactly (suppl., p25). Next, we want to bound the error and give an estimate of it. In contrast, Dropout only tries to make the error smaller.
>
> [1] P. Kerlirzin and F. Vallet. Robustness in multilayer perceptrons
> [2] G. E. Hinton, N. Srivastava, A. Krizhevsky, I. Sutskever, and R. R. Salakhutdinov. Improving neural networks by preventing co-adaptation of feature detectors
> [3] A. Krizhevsky, I. Sutskever, and G. E. Hinton. Imagenet classification with deep convolutional neural networks
>
> (clarity: Dropout and Fast Dropout) Dropout is a regularization technique using randomly crashing neurons during training. We study fault tolerance, which is a different problem: be resilient at the stage of inference to these randomly crashing neurons. We set our goal to calculate the error in the output. Dropout theory is aimed at explaining generalization properties and providing faster versions of Dropout. In particular, the paper on Fast Dropout considers the first-order Taylor expansion without bounding the remainder, as no guarantee is required. While we also use a Taylor expansion, in contrast, we explicitly bound the second-order remainder to guarantee robustness in all cases.
>
> We note again that we solve a different problem compared to the study of Dropout.
>
> (Adversarial Examples and Dropout) These directions are interesting. However, we do not address the worst case of adversarial examples and we consider the average case. This would be an interesting research direction, but out of the scope of this paper.
>
> (Motivation of the paper) We have created a website briefly describing the paper's motivation, positioning and the approach: https://iclr-2020-fault-tolerance.github.io . Below we also describe the major points.
>
> Our motivation is the following: emergent neuromorphic hardware is efficient but prone to faults. We study the effect of such faults on the computation by bounding the error theoretically, and then show how to defend against them using a regularizer. Our problem that we consider (Fault Tolerance) is related to Dropout, but is distinct from Dropout.
>
> (Biological fault tolerance and squids) There is no known neurons in complex organisms where a single neuron loss would lead to a remarkable effect. Any organism whose survival or any major function would rely on a single neuron would have been cleared by the evolution rapidly.
>
> In general, a human can sustain damage to entire regions of the brain (thousands of neurons) without loosing too much capacities. For example, a person had a nail in their brain for hours without realizing it [1]. In another case, the person had 90% of his brain slowly cease to exist, without any symptoms [2].
>
> From an evolutionary point of view, single cell die quite easily -- just because neural activity is extremely cytotoxic [3]. For instance visual cortex pyramidal neurons need to be aneuploid [4,5] just to sustain the stress induced by their activity. For example, for C. elegans’s it was found that the neural system rewires itself under stress, thus it is fault-tolerant [6].
>
> [1] https://www.theguardian.com/world/2012/jan/21/man-survives-shooting-nail-brain
> [2] https://www.sciencealert.com/a-man-who-lives-without-90-of-his-brain-is-challenging-our-understanding-of-consciousness
> [3] https://link.springer.com/article/10.1007/s12640-009-9051-z
> [4] https://www.pnas.org/content/102/17/6143
> [5] https://www.ncbi.nlm.nih.gov/pmc/articles/PMC5490080/
> [6] https://www.sciencedirect.com/science/article/pii/S0092867418316386
>
> (Speed of Light and Neuromorphic hardware) Electrical signals are transmitted to the speed close to the speed of light.  Depending on the parameter of material called velocity factor that speed can range from 50% to 99% speed of light [1]. Compared to the current load-offload cycle of thousands nanoseconds (thousands of instructions, at a clock frequency of ~1GHz) in best of conditions of CPUs/GPU for a single propagation step, the speed of light computation is exponentially faster.
> [1] https://en.wikipedia.org/wiki/Speed_of_electricity
>
> (content: Citation format) Thank you for the note, we make the style compatible with ICLR in the final version.
>
> Do not hesitate to ask us for more clarification, if that is needed.

---

### Official Review · AnonReviewer3 · 2019-10-24
**Official Blind Review #3**

**Rating:** 3

**Review:**

This contribution studies the impact of deletions of random neurons on prediction accuracy of trained architecture, with the application to failure analysis and the specific context of neuromorphic hardware. The manuscript shows that worst-case analysis of failure modes is NP hard and contributes a theoretical analysis of the average case impact of random perturbations with Bernouilli noise on prediction accuracy, as well as a training algorithm based on aggregation. The difficulty of tight bounds comes from the fact that with many layers a neural network can have a very large Lipschitz constant. The average case analysis is based on wide neural networks and an assumption of a form of smoothness in the values of hidden units as the width increases. The improve fitting procedure is done by adding a set of regularizing terms, including regularizing the spectral norm of the layers.

The robustness properties can be interesting to a wider community than that of neuromorphic hardware. In this sense, the manuscript provides interesting content, although I do fear that it is not framed properly. Indeed, the introduction and conclusion mention robustness as a central concern, which it is indeed, but the neuron failures are quite minor in this respect. More relevant questions would be: is the approach introduced here useful to limit the impact of adversarial examples? Does it provide good regularizations that improve generalization? I would believe that the regularization provided are interesting by looking at their expression; in particular the regularization of the operator norm of layers makes a lot of sense. That said, there is some empirical evidence that batch norm achieves similar effects, but with a significantly reduced cost.

Another limitation of the work is that it pushes towards very wide networks and ensemble predictions. These significantly increase the prediction cost and are often frowned upon by applications.

It seems to me that the manuscript has readability issues: the procedure introduced is quite unclear and could not be reimplemented from reading the manuscript (including the supplementary materials). Also, the results in the main part of the manuscript are presented to tersely: I do not understand where in table 2 dropout is varied.

The contributed algorithm has many clauses to tune dynamically the behavior of the regularizations and the architecture. These are very hard to control in theory. They would need strong empirical validation on many different datasets.

It is also very costly, as it involves repeatedly training from scratch a neural network.

The manuscript discusses in several places a median aggregation, which gives robustness properties to the predictor. I must admit that I have not been available to see it in the algorithm. This worries me, because it reveals that I do not understand the approach. The beginning of section 6.1, in the appendix, suggests details that are not understandable from the algorithm description.

Finally, a discussion of the links to dropout would be interesting: both in practice, as dropout can be seen as simulating neuron failure during training, as well as from the point of view of theory, as there has been many attempts to analyze theoretically dropout (starting with Wager NIPS 2013, but more advanced work is found in Gal ICML 2015, Helmbold JMLR 2015, Mianjy ICML 2018).

**Experience Assessment:**

I have published one or two papers in this area.

**Review Assessment: Checking Correctness Of Derivations And Theory:**

I assessed the sensibility of the derivations and theory.

**Review Assessment: Checking Correctness Of Experiments:**

I assessed the sensibility of the experiments.

**Review Assessment: Thoroughness In Paper Reading:**

I read the paper thoroughly.

---

> ### Author Response · Authors · 2019-11-12
> **We explain the paper positioning better, run additional experiments and explain unclear parts**
>
> Dear Reviewer, thank you for the review.
>
> We provide the detailed description of the motivation of the paper here: https://iclr-2020-fault-tolerance.github.io/
>
> Weight matrices norm was proposed as regularization before [1]. Our novelty is computing the error in the continuous limit (Th1). Adversarial examples are the worst-case perturbations. We, in contrast, consider the average case. We only tangentially discuss them on page 18 of the suppl.
> [1] Gouk, Henry, et al. "Regularisation of neural networks by enforcing lipschitz continuity."
>
> Dropout is indeed connected to Fault Tolerance. It is in this context that this algorithm was invented [1]. Reducing overfitting came out as a bonus in Kerlirzin’s work and Dropout would, two decades later, be rebranded [2] as a regularization method. We, however, are interested in calculating the error rather than in generalization properties. While Dropout helps fault tolerance, it is not a solution to our problem: first, we do not know the $p$ exactly (suppl., p25). Next, we want to estimate the error, and not just make the error smaller.  We have added a paragraph summarizing this difference to the introduction in the final version.
>
> [1] P. Kerlirzin and F. Vallet. Robustness in multilayer perceptrons
> [2] G. E. Hinton et al Improving neural networks by preventing co-adaptation of feature detectors
>
> As our Conclusion mentions, there are links between generalization and fault tolerance. While exploring the connection to the generalization problems would be exciting, as we mention in the conclusion, it is out of the scope of this paper.
>
> The major solution to the problem of fault tolerance is redundancy [1]. The obvious way to apply that to neural networks is to replicate an off-the-shelf network multiple times. However, the single network itself might not utilize its neurons efficiently for fault tolerance. We improve the robustness of a single network.
>
> Batch normalization for improving fault tolerance is an interesting idea, but not a well-studied one. We run an experiment to determine the effect of batch normalization on fault tolerance. It shows that this technique does not result in major improvements for a small MNIST CNN: https://cutt.ly/0ePOco1
> [1] von Neumann, J. (1956). "Probabilistic Logics and Synthesis of Reliable Organisms from Unreliable Components"
>
> Indeed, table 2 is confusing. We have adjusted it in the final revision of the paper. Now we provide an explanation for the experiment and the table.
>
> The goal is to compare fault tolerances of different networks under faults with prob. $p$. We take several train Dropout parameters $p_{train}$ and train the networks. We train with each $p_{train}$ 6 times, doing more repetitions until the variance is low enough.
>
> It is known that when increasing $p_{train}\in [0, p]$, the fault tolerance increases. This is also true in our experiment: the Experimental part  of Table 2 shows that crashing mean absolute error (MAE) "correlates" well with $p_{train}$ (as via rank loss).
>
> We are looking for a metric which can order correctly which network is more resilient. We take crashing (for the network with faults)/correct (without faults) accuracy and MAE for test/train datasets, and also our bounds. We compute the rank loss between the metric and $p_{train}$. This shows how well the metric "correlates" with the resilience to faults. We only care about the correct ordering, so we use rank loss instead of correlation (more in suppl., Sec, 7.1-7.2).
>
> Two parts of the table (left and right) have the same structure, and just describe different metrics. The alignment in the table design does not indicate that theoretical values should be compared to the experimental ones.
>
> The table shows that only the T1 VarDelta bound has a low rank loss. This means that T1 can tell which network is more fault-tolerant, unlike the other metrics.
>
> The algorithm lacks clarity in a sense that some constants are not explained due to the paper to the size limit. We made it more clear in the final version. Here we provide explanations on the algorithm.
> (line 6) Constants $1/3$ and $\alpha$ are chosen in supl., Section 6.1
> (line 7) $n$ is the number of neurons at a layer with crashes
> (line 8) $R_3$ is defined below Eq. 1
> (line 9) The constant $C$, complexity, is fully defined in suppl, p18
> (line 12, 13) $n$ is the number of neurons at the layer with crashes
> (line 14) We copy the network multiple times $R$ (in hardware) with median aggregatio. We do not implement is as it is a well-established technique
>
> Our algorithm is a proof-of-concept, and it is not a major feature of the paper. In contrast, our bound (Theorem 1) is. Using it, we only show that our theoretical bound is also practical.
>
> We consider neural networks implemented in neuromorphic hardware as an application. Thus, it is less important how expensive is the training procedure since it is done only once.
>
> Do not hesitate to write a response to us if you have more questions.

---

### Official Review · AnonReviewer2 · 2019-10-26
**Official Blind Review #2**

**Rating:** 8

**Review:**

This paper investigates the problem of fault telorance
on NN: basically how the predictions are affected by
failure of certain neurons at prediction time. The analysis
is theoretical under the classical assumption of lipschitz
bounded non-linearities and looking at the limit case
of an infinite number of neurons. The paper is well
written and comes with public code that allows to replicate
the experiments illustrating the theoretical derivations.

The paper is well motivated and addresses a timely matter.

Typos

- "the error of the output of" -> "the error of the output"


**Experience Assessment:**

I do not know much about this area.

**Review Assessment: Checking Correctness Of Derivations And Theory:**

I assessed the sensibility of the derivations and theory.

**Review Assessment: Checking Correctness Of Experiments:**

I assessed the sensibility of the experiments.

**Review Assessment: Thoroughness In Paper Reading:**

I made a quick assessment of this paper.

---

> ### Author Response · Authors · 2019-11-12
> **Thank you for your review**
>
> Dear Reviewer,
>
> Thank you for your time. We thank you for your review. The error is corrected in the final version. We will remain at your disposal for any additional clarifications, if the need for them to arise for you.

---

### Decision · Program_Chairs · 2019-12-19

**Decision:**

Reject

**Comment:**

 This paper focuses on the problem of robustness in the network with random loss of neurons.  However, reviewers had issues with insufficient clarity of the presentation, and lack of discussion about closely related dropout approach.